# SLE serum induces altered goblet cell differentiation and leakiness in human intestinal organoids

Inga Viktoria Hensel [1], Szabolcs Éliás[1], Michelle Steinhauer[1], Bilgenaz Stoll [1], Salvatore Benfatto [1], Wolfgang Merkt[2], Stefan Krienke[2], Hanns-Martin Lorenz [2], Jürgen Haas[3], Brigitte Wildemann[3] & Martin Resnik-Docampo [1]✉

## Abstract

**Human intestinal epithelial cells are the interface between luminal content and basally residing immune cells. They form a tight monolayer that constantly secretes mucus creating a multilayered protective barrier. Alterations in this barrier can lead to increased permeability which is common in systemic lupus erythematosus (SLE) patients. However, it remains unexplored how the barrier is affected. Here, we present an in vitro model specifically designed to examine the effects of SLE on epithelial cells. We utilize human colon organoids that are stimulated with serum from SLE patients. Combining transcriptomic with functional analyses revealed that SLE serum induced an expression profile marked by a reduction of goblet cell markers and changed mucus composition. In addition, organoids exhibited imbalanced cellular composition along with enhanced permeability, altered mitochondrial function, and an interferon gene signature. Similarly, transcriptomic analysis of SLE colon biopsies revealed a downregulation of secretory markers. Our work uncovers a crucial connection between SLE and intestinal homeostasis that might be promoted in vivo through the blood, offering insights into the causal connection of barrier dysfunction and autoimmune diseases.**

**Keywords** Systemic Lupus Erythematosus (SLE); Gut Leakiness; Organoids; scRNA-seq; Intestinal Homeostasis
**Subject Categories** Digestive System; Immunology; Methods & Resources

## Introduction

The intestine has a substantial surface area that harbors the highest quantity of immune cells in close proximity to the tremendous number of microbes in our body (Mowat and Agace, 2014; Knoop and Newberry, 2018; Helander and Fändriks, 2014). Thus, a functional epithelial barrier is crucial for separating these two compartments to ensure intestinal homeostasis and overall health. Under physiological conditions this function is maintained by constant renewal of the barrier containing a balanced cellular composition (Horowitz et al, 2023; Beumer and Clevers, 2021; Barker et al, 2010). Intestinal stem cells and transit-amplifying (TA) cells proliferate and give rise to different specialized cells committed to either absorptive or secretory lineage, represented by the two major cell types: colonocytes and goblet cells. While colonocytes mainly contribute to maintaining fluid balance and absorbing nutrients, goblet cells are considered as key players of mucosal barrier integrity. The secreted mucus forms a layer which is not only essential as defense against microbial infiltration, but also acts as a niche for commensals (Gehart and Clevers, 2019; Dutton et al, 2019; Beumer and Clevers, 2021; Allaire et al, 2018). Its composition is highly dynamic, can be influenced not only by the abundance of mucus core proteins and antimicrobial peptides secreted to it but also by factors such as ion concentration, pH, and hydration state (Hansson, 2019; Pelaseyed et al, 2014). The cell-type composition and therewith barrier function is continuously influenced by signals from neighboring immune and stromal cells, cytokines, bacterial metabolites, and nutrients. Hallmarks of pathological conditions like inflammatory bowel disease and infections are shifts in cell-type proportions and alterations in mucus composition affecting barrier function and increasing intestinal permeability (Dotti et al, 2017; Van Der Post et al, 2019). Recent research advancements have started to shed light on the broader implications of gut leakiness. The discovery that the intestine plays a considerable role in the pathogenesis of a number of extraintestinal systemic diseases marks a paradigm shift, expanding our understanding of the pivotal role the intestine plays in overall health and disease (Mu et al, 2017; Azzouz et al, 2019).

Systemic lupus erythematosus (SLE) is a multifaceted autoimmune disease which is characterized by systemic inflammation affecting the skin, kidney, and the central nervous system (Ruiz-Irastorza et al, 2001). Recent reports show that SLE patients have an altered microbiome (Zhang et al, 2014; He et al, 2016; Hevia et al, 2014; Li et al, 2019; Luo et al, 2018; Singh et al, 2017) and increased intestinal permeability (Azzouz et al, 2019; Shi et al, 2014; Issara-Amphorn et al, 2018; Silverman et al, 2019; Nockher et al, 2008). Limited knowledge exists regarding the role and contribution of the intestine in the pathogenesis and progression of the disease. Especially in which extent the intestinal epithelial cells are affected remains to be elucidated. SLE manifests with a dysregulation of the immune system and is characterized by elevated

[1]BioMed X Institute, Heidelberg, Germany. [2]Division of Rheumatology, Department of Medicine V, University Hospital Heidelberg, Heidelberg, Germany. [3]Molecular Neuroimmunology Group, Department of Neurology, University of Heidelberg, Heidelberg, Germany. ✉E-mail: resnik@bio.mx

type I interferon levels and the presence of autoantibodies in serum (Yao et al, 2010; Davis et al, 2011; Becker et al, 2019; Fasano et al, 2023). The blood is a complex body fluid that serves as transport medium to supply all cells of the body with nutrients and oxygen. The intestine is the most intensely perfused organ. Especially the metabolically highly active epithelial cells are in close contact with the underlying vasculature, ensuring swift exchange and efficient nutrient absorption necessary for homeostasis (Matheson et al, 2000). However, this may also facilitate an influx of circulating cytokines and autoantibodies contained in the blood to the intestine with the potential to alter epithelial cell dynamics and barrier function (Danahay et al, 2015; Katlinskaya et al, 2016a; Simões et al, 2021). A comprehensive understanding of this interconnection is of great importance to elucidate the role of the intestinal epithelial barrier in autoimmune diseases and for the development of novel therapeutic approaches.

To fully understand this complex interplay in a multicellular environment, it is necessary to develop a model that can explore the impact of blood components on the intestinal epithelium in near-physiological conditions. This model, however, should be strategically designed to only include intestinal epithelial cells. By specifically focusing on these cells, potential interference from other components on opposite sides of the barrier can be eliminated. This includes microbiota on the luminal side, as well as immune and stromal cells on the basolateral side. In this way, direct effects of blood components on intestinal epithelial cells can be isolated and observed, providing a more controlled and precise understanding of these interactions. To this end, the development of organoids as advanced primary cell model has revolutionized the design of human tissue models. These self-organized, three-dimensional structures, derived from adult stem cells, surpass traditional cell models while recapitulating the organ architecture and plasticity found in vivo (Fujii et al, 2018). Organoids have become a notable alternative to animal models, offering a more reliable and human-relevant approach for addressing long-standing medical challenges. The relevance of organoids as a model system has been recognized by the FDA that recently approved it to be used as a non-animal model in the drug development process (Wadman, 2023). This marks the beginning of a new era in biomedical research. Our research takes a pioneering approach by integrating serum derived from SLE patients into intestinal organoids enabling us to simulate the impact of blood components on the intestinal epithelium while minimizing the interference caused by the complex intestinal microenvironment, providing a focused, more controlled environment for our explorations and analyses.

Herein, we report that our organoid model contains all relevant epithelial cell types found in the colon. We show that SLE serum stimulation can induce alterations in the expression profile, which are dependent on type I interferon signaling, induced through a synergistic effect of all serum-contained factors and are specific to SLE. Our results demonstrate increased barrier permeability and a significant change in the secretory lineage, potentially leading to a weakened mucus layer due to alterations in its composition. Conclusively, our in vitro model emulates a pathological condition where a complex mix of cytokines and other SLE-specific factors secreted at the site of inflammation and distributed systemically reaching the highly perfused intestine via the blood can impact the epithelial barrier. This innovative disease model holds the potential to unveil unique insights into disease mechanisms that would otherwise be challenging to investigate. In addition, it holds the potential as a tool to identify innovative treatment approaches and points of intervention.

# Results

## Organoids show SLE-specific response to serum stimulation

To understand the effect of SLE on the intestinal epithelial barrier, we generated organoid line I and II from the descending colon of two healthy donors, cultured them in conditions that conserved the native cellular diversity (Fujii et al, 2018), and stimulated them for 72 h with 5% serum from treatment-naive SLE patients or sex- and age-matched controls (Fig. 1A; Table EV1). Serum stimulation did not induce any significant phenotypical changes in size or shape (Fig. 1B). There was no evidence for increased cytotoxicity or apoptosis (Figs. 1C and EV1A). Expression profiles of the stimulated organoids were marked by a distinct spread in expression profiles for both stimulation conditions across two tested organoid lines (Fig. 1D). The first principal component (PC1), accounting for 39% of the total variance, displayed a separation of a subgroup of organoids stimulated with SLE sera from the controls in organoid line I. Interestingly, the same SLE sera induced an even more pronounced separation in organoid line II along the PC1 axis. This consistent separation across both organoid lines strongly suggested a donor-independent effect of SLE serum stimulation on epithelial cells. The second principal component (PC2), accounting for 15% or 21% of the total variance, further differentiated the samples within each condition, likely reflecting serum sample-specific responses.

Taken together, these results reinforced the inherent complexity of serum-stimulated organoids and the need to consider both stimulation- and organoid donor-specific effects for downstream analysis and data interpretation. Therefore, we pooled the data from both organoid donors accounting for sex and individual characteristics of the organoid line. In addition, we integrated the response of different cell types given the fact that the cell-type composition in both organoid lines varied due to individual proliferation and differentiation dynamics (Fig. EV1B). The combined analysis resulted in 256 differentially expressed genes (Fig. 1E). Gene Ontology (GO) analysis of the upregulated genes showed an overrepresentation of terms connected to cell cycle, chromosome organization and replication as well as mitochondrial function and interferon signaling (Fig. 1F). The downregulated genes showed an enrichment of terms connected to secretion, cytoskeleton, and anchoring junctions of the cells (Fig. 1G), suggesting changes in barrier function. We therefore performed a permeability assay which revealed increased paracellular transport of fluorescein isothiocyanate (FITC) of organoid monolayers upon 72 h exposure to SLE compared to control serum (Fig. 1H).

Intriguingly, the overall results showed that similar pathways were altered in epithelial cells as previously only described in immune cells of SLE patients (Zhang et al, 2021). Thus, the distinct response of organoids to SLE patient-derived serum, consistent in both organoid lines, holds the potential to unravel how SLE-specific mechanisms can influence the intestinal epithelial barrier. Furthermore, the changes seen on the expression level were translated into functional changes as seen by an increased paracellular flux upon SLE serum stimulation, indicating barrier leakiness.

## Type I interferon drives the expression changes induced by SLE serum

We wanted to explore the overrepresentation of terms connected to interferon (IFN) signaling more in-depth. Out of the 256 DEG, we

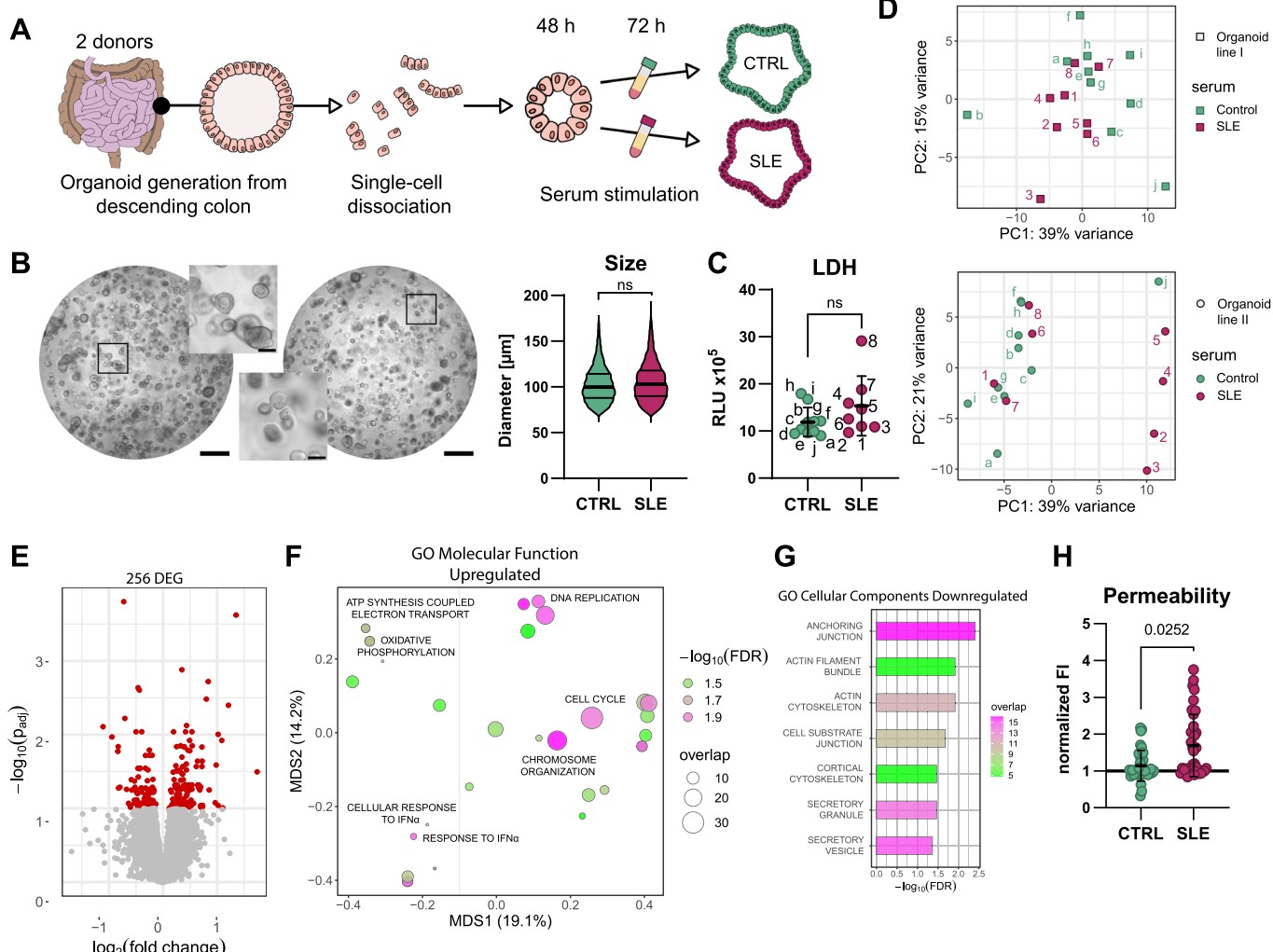

**Figure 1. Transcriptomic profiling of serum-stimulated organoids reveals SLE-specific responses.**

(A) Schematic representation of the experimental approach. (B) Bright-field microscopy images of colonic organoids after 72 h stimulation with either control (left) or SLE serum (right). Insets showcase enlarged areas; scale bars: 1000 μm and 100 μm, respectively. The graph (right) shows the analysis of the organoid size comparing SLE to control serum-stimulated organoids. Data is presented as violin plot with lines at median and quartiles; statistical significance was determined using chi-square test considering 100 averaged values of 10,000 organoids in total. (C) Cytotoxicity analysis using lactate dehydrogenase (LDH) release comparing SLE to control serum-stimulated organoids. Data are presented as mean with ±SD with each dot corresponding to one well of serum-stimulated organoids. Statistical significance was determined by the Kolmogorov–Smirnov test. (D) PCA analysis was performed on organoids stimulated for 72 h with control (green) and SLE (pink) serum. Upper panel shows the results from organoid line I and lower panel from organoid line II. (E) Volcano Plot showing the 256 DEGs (red) comparing organoids stimulated with SLE or control serum for 72 h. (F) Multidimensional scaling (MDS) plot depicting differently enriched gene sets of upregulated genes according to the GO term "Molecular Function" comparing SLE to control serum-stimulated organoids. (G) Table showing differently enriched gene sets of downregulated genes according to the GO term "Cellular Components" comparing SLE to control serum-stimulated organoids. (H) Analysis of barrier permeability of organoid monolayers stimulated for 72 h with control (green) or SLE (pink) serum using FITC. Each dot represents one well of serum-stimulated organoids ($n = 8$ control or SLE sera, respectively; $n = 3–5$ technical replicates; two independent experiments). Data are represented as mean with ±SD, statistical significance as determined by Kolmogorov–Smirnov test. Unless otherwise specified, all experiments depicted in this figure were analyzed with $n = 10$ control serum (samples a–j) and $n = 8$ SLE serum (samples 1–8). Except in (C, H) both organoid lines I and II were analyzed. Source data are available online for this figure.

found 22 genes to be connected to type I IFN signaling (Fig. 2A). A significant majority of them were upregulated. They accounted for 32% of the 25 highest upregulated genes, underlining the role of type I IFN in the response induced by SLE serum stimulation. To investigate whether the expression of IFN-related genes was altered in both organoid lines by the different patient sera, we selected a range of IFN-inducible genes that were represented in several GO terms (Fig. 2B). The representation of their expression in a heatmap

revealed an upregulation in the majority of SLE serum-stimulated organoids indicating the SLE specificity of this response (Figs. 2B and EV1C; Dataset EV1).

A common way to analyze the type I IFN levels and activity in SLE serum samples is to analyze the expression profile of interferon signature genes (IFNSG) in SLE whole blood samples (Brohawn et al, 2019). Given the fact that we were studying epithelial cells we decided to choose the IFNSG based on IFN-α stimulated organoids.

Considering only genes relevant in IFN-α/β signaling, we identified 27 genes which were specific for epithelial cell response to a low dose of IFN-α (Fig. EV1D). We used this panel of genes (irrespective of their significance of expression) to calculate an IFN score from the normalized counts of SLE and control serum-stimulated organoids (Fig. 2C). We could see an overall higher score in organoids stimulated with SLE serum compared to control serum irrespective of the organoid line. The IFN score gave us a powerful tool to assess the expression of IFNSG unique to each serum independent of the significance in expression identified by the pooled data. Thus, we were able to integrate and interpret the effect by IFN in a more robust way than just focusing on single gene expression or DEGs which are based on the mean expression. In addition, we ruled out that the seen IFN signature was caused by endogenous expressed IFN by confirming the absence of IFN-α1 and IFN-β1 transcripts (Fig. EV1E). We could confirm the activation of IFN signaling by upregulation and phosphorylation of STAT1 which was SLE serum concentration-dependent and almost absent in control serum-stimulated organoids (Figs. 2D and EV1F). Only with SLE serum or IFN-α stimulation, we could detect IFIT3, interferon-induced protein with tetratricopeptide repeat-3, abundance (Fig. EV1F).

As a next step, we wanted to quantify type I IFN levels of the used serum samples. Using a bead-based immunoassay to detect IFN-α2 levels, we could only detect a minor increase in SLE serum compared to the control (Fig. 2E). However, this is in line with literature reporting challenges in the detection of IFN-α2 in serum (da Silva et al, 2021; Rodero et al, 2017). Therefore, we employed a second approach stimulating an IFN-α/β reporter cell line with the serum samples. With this functional readout, we were able to detect a significant increase in type I IFN levels in SLE serum (Fig. 2F). To better understand the effects of the serum on the organoids, we also checked several other cytokines. We could observe the significantly elevated level of IL-18 and the presence of IL-6 in almost all SLE, but not control serum samples (Fig. 2G), as it has been reported for bigger SLE cohorts (Ohl and Tenbrock, 2011; Guimarães et al, 2017; Mende et al, 2018; Kulkarni et al, 2007; Živković et al, 2018). Even though some of the control sera also showed increased levels of single cytokines, only the SLE serum samples showed an overall increase of several cytokines (Fig. 2H). This result did not only indicate the overall inflammatory signature in SLE serum samples, but also showed the diversity of the applied samples.

Taken together, our data revealed that organoids stimulated with serum derived from SLE patients were characterized by type I IFN signature similar to that of patient-derived immune cells. The activation of IFN signaling underlined the so far neglected impact of IFN on intestinal epithelial cells in the context of SLE.

## SLE serum stimulation leads to altered mitochondrial function

We wanted to understand which other effects SLE serum would have on epithelial cells. The term mitochondrion represented 10% of all DEGs (Fig. 3A), while the enrichment analysis showed an upregulation of genes relevant for oxidative phosphorylation (Fig. 1F). Mitochondria are implicated in SLE pathogenesis (Buang et al, 2021; Zhao et al, 2022) and serve as crucial regulators in maintaining intestinal homeostasis (Crakes et al, 2019; Khaloian et al, 2020). Closer analysis revealed that mitochondrially encoded genes of complex I and IV exhibited high expression levels in general (Figs. 3A,B and EV2A). Thus, their further upregulation in SLE-stimulated organoids indicated their potential impact on respiratory chain activity. Evaluation of expression levels for each SLE serum donor demonstrated that this effect was induced by almost all samples (Figs. 3B and EV2A). To exclude that the overall upregulation of mitochondrial-related genes was caused by an increase in mitochondrial mass, we assessed the mitochondrial content in the serum-stimulated organoids. Since it is known that the mitochondrial morphology is cell-type-dependent (Stringari et al, 2012) we decided to use flow cytometric analysis to quantify overall changes in mitochondrial mass. The results revealed that there were no significant differences in mitochondrial mass between both experimental conditions (Figs. 3C and EV2B). This indicated that the upregulated mitochondrial genes were due to an increased mitochondrial activity rather than caused by an increase of mitochondrial mass.

We assessed mitochondrial function by a respiratory assay and saw a significant increase of basal respiration and ATP production along with an unaltered maximal respiration upon SLE serum stimulation (Figs. 3D and EV2C). Similar results were seen when CD8[+] T cells were stimulated with IFN-α (Buang et al, 2021). Upon assessing the relative spare respiratory capacity, a notable decrease was observed following SLE serum stimulation, which aligned with previous reports on altered mitochondrial function in CD8[+] T cells derived from SLE patients (Buang et al, 2021). These results indicated that we were looking at an increased basal energy production and a diminished capacity of intestinal epithelial cells to adjust to a dynamic energy demand after SLE serum stimulation.

Overall, we could see that the metabolic profile of the organoids was altered upon SLE serum stimulation (Fig. 3E). We were interested in understanding if this alteration was caused by a changed cell-type composition of the organoids. Reports from literature suggest that more differentiated cells switch from glycolysis to oxidative phosphorylation as their primary energy source which would be reflected by a shift from the right lower to the left upper quadrant in the energy map (Stringari et al, 2012; Guerbette et al, 2022; Rath et al, 2018; Ludikhuize et al, 2020). However, there is limited knowledge about the metabolic profile of different cell types in the human colon. We therefore generated organoids with different cell-type composition and observed a metabolic shift with a trend towards higher oxidative phosphorylation upon differentiation (Fig. EV2D,E). The metabolic shift we observed upon SLE serum differed and showed a trend to higher glycolytic activity compared to the control (Fig. 3E). While the impact of differentiation primarily relied on OCR (oxygen consumption rate), the effects of SLE stimulation influenced mainly ECAR (extracellular acidification rate). These results suggested that the changes induced by SLE serum stimulation had a more complex cause than those induced by differentiation and might be additionally caused by an overall change in mitochondrial function.

## Expression of secretory cell markers are reduced, while proliferation seems to be unaffected

Our in vitro model offered a unique opportunity to examine whether serum stimulation could affect differentiation dynamics, thereby replicating the potential impact on the continuous

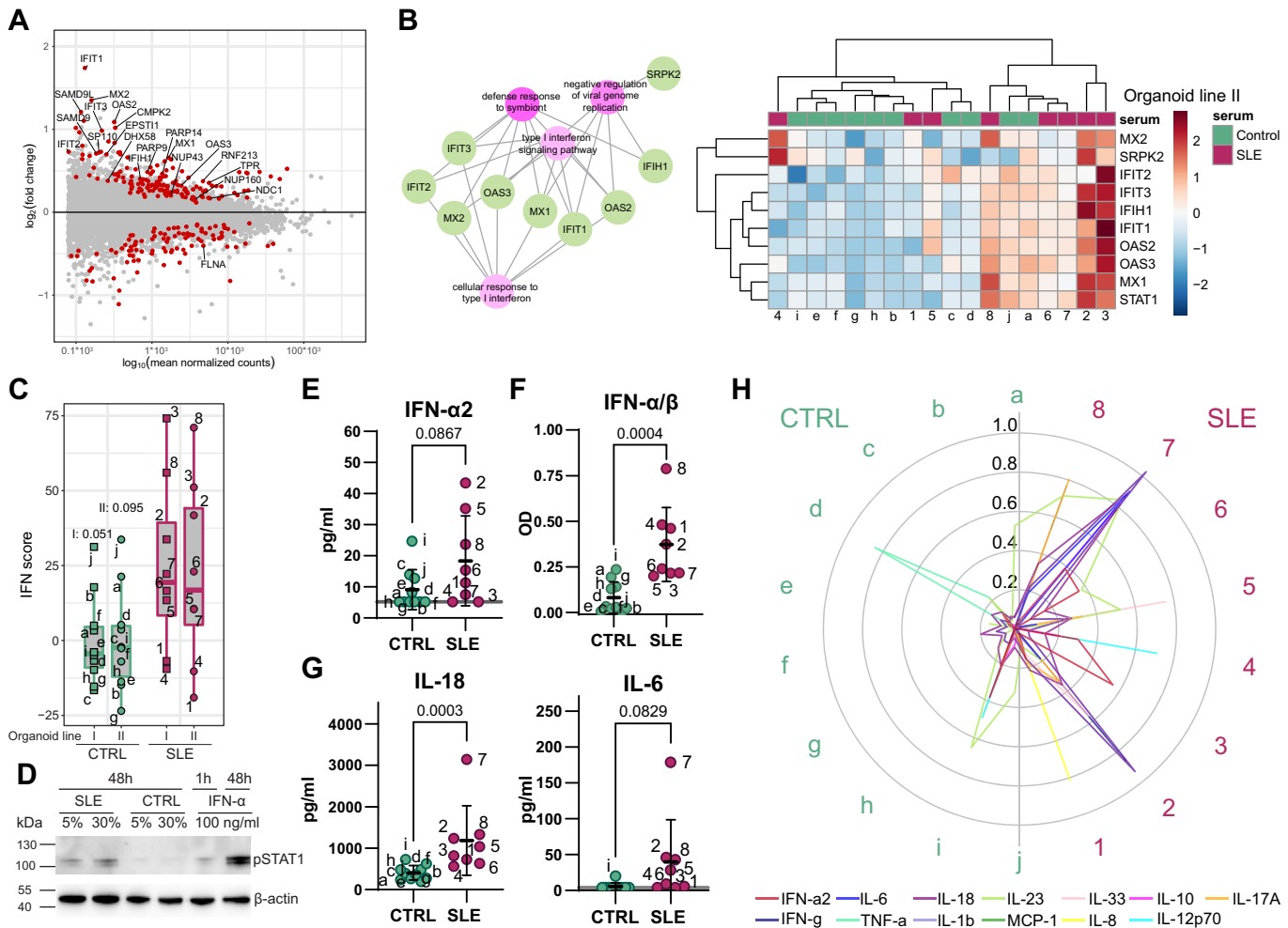

**Figure 2. Organoid expression profile induced by complex cytokine composition in SLE serum is IFN-α driven.**

(A) MA plot showing DEGs (red) and genes associated with IFN signaling highlighted by the gene name label comparing SLE to control serum-stimulated organoids. (B) Netplots highlighting IFN-inducible genes and their corresponding gene sets (left). Heatmap with hierarchical clustering of selected IFN-inducible genes comparing SLE ($n = 8$) to control serum ($n = 10$) stimulated organoids from organoid line II. (C) Analysis of the IFN score calculated as z-scores of normalized counts of selected IFN-inducible genes. Data are represented as mean with ±SD (box) and 95% confidence interval (whisker). Statistical differences as determined by one-way ANOVA with Holm–Šidák's multiple comparisons test. (D) Western blot analysis of STAT1 phosphorylation in IFN-α, control and SLE serum-stimulated organoids after 48 h. Each sample represents two pooled wells of organoids stimulated with serum from one donor. (E) Multiplex ELISA analysis showing IFN-α2 levels of control (green) and SLE serum (pink). Data are represented as mean ± SD, and statistical significance determined by unpaired t test. (F) Analysis of type I interferon serum levels using an IFN-α/β reporter cell line stimulated with control (green) or SLE serum (pink). Optical density (OD) was assessed for analysis. Data are represented as mean ± SD, and statistical significance determined by the Kolmogorov–Smirnov test. (G) Multiplex ELISA analysis showing IL-18 and IL-6 levels of control (green) and SLE serum (pink). Data are represented as mean ± SD, and statistical significance determined by unpaired t test or Mann–Whitney test, respectively. (H) Radar plot illustrating the distinct cytokine composition in each serum sample, quantified by multiplex ELISA. Each line color represents one cytokine, levels are shown as fraction, and normalized to the highest and lowest value for each cytokine. Green and pink labels represent control and SLE samples, respectively. All experiments depicted in this figure were analyzed with $n = 10$ control serum (samples a–j) and $n = 8$ SLE serum (samples 1–8). (A–C) Pooled results of organoid line I and II are shown. (E, G) The gray line shows ½ the detection limit which was used for statistical analysis if the cytokine was undetectable. Source data are available online for this figure.

regeneration of the epithelial cell layer in vivo during SLE onset and progression. Cytokines have the capacity to shape the proliferation and differentiation dynamics (Major et al, 2020; Katlinskaya et al, 2016b). However, not much is known about how epithelial cells specifically are influenced by cytokines at in vivo relevant concentrations contained in the serum of SLE patients. Analysis of the proliferative capacity of organoids stimulated with SLE serum did not show any changes when size, cell cycle (KI67), or S phase (EdU) was analyzed (Figs. 1B and EV3A,B). Both our lines showed slightly different responses, most likely due to given

differences in individual differentiation dynamics (Fig. EV1B). However, the hyperenrichment analysis considering both lines revealed the terms secretory vesicle and secretory granule amongst the top 7 downregulated gene sets (Figs. 1G, 4A, and EV3C). This indicated an effect on the secretory function of the epithelial barrier, so we focused on cell-type composition changes in the organoids. We checked a panel of cell-type marker genes which showed almost exclusively effects on differentiated cells as indicated by a decrease of absorptive and secretory lineage markers (Fig. 4B). Of note, the absorptive cell markers SERPINA1, Serpin Family A

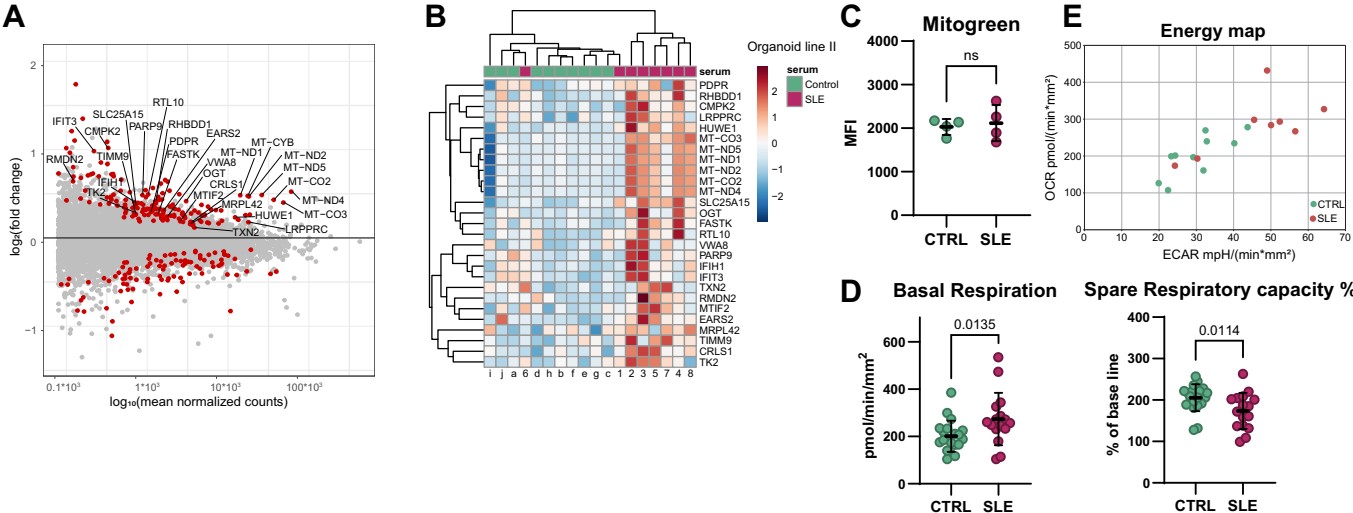

**Figure 3. SLE serum stimulation leads to altered mitochondrial function.**

(A) MA plot showing DEGs (red) and genes associated with the gene set "Mitochondrion" highlighted by the gene name label comparing SLE to control serum-stimulated organoids. (B) Heatmap with hierarchical clustering of mitochondrial genes comparing SLE to control serum-stimulated organoids from organoid line II. (C) Flow cytometry analysis of SLE and control serum-stimulated organoids stained with mitogreen. Each dot represents one well of organoids dissociated into single cells and analyzed as mean fluorescence intensity (MFI); $n = 4$ serum for each condition. Data are represented as mean ± SD, and statistical significance determined by unpaired $t$ test. (D) Seahorse assay showing basal respiration (left) and relative spare respiratory capacity (right) comparing SLE (pink) to control (green) serum-stimulated organoids. Data are represented as mean ± SD, and statistical significance determined by unpaired $t$ test. (E) Energy map illustrating the metabolic profile of organoids following stimulation with control (green) or SLE serum (pink). (A, B) Pooled results of organoid line I and II are shown. (D, E) Each dot corresponds to one well of organoids stimulated with $n = 10$ control serum or $n = 8$ SLE serum samples. Each serum stimulation was analyzed with $n = 2$ technical replicates. Source data are available online for this figure.

Member 1 (also known as AAT), and SCNN1A, Sodium Channel Epithelial 1 Subunit Alpha (also known as ENaC), are both connected to mucus layer build-up and function. SERPINA1 has antimicrobial functions (Janciauskiene et al, 2011), while SCNN1A is important for ion and fluid regulation (Baker et al, 2012) thereby playing a role in modifying mucus characteristics. Amongst the secretory cell markers were well-known goblet cell markers AGR2, TFF3, and SPINK4 as well as CHGA which is a marker for enteroendocrine cells. AGR2 is essential for the production and processing of gel-forming mucins such as MUC2 (Al-Shaibi et al, 2021), whereas TFF3 forms a complex with FCGBP, one of the main components of mucus (Albert et al, 2010; Ehrencrona et al, 2021). We were therefore especially interested to understand if we could detect changes in the secretory lineage. The number of goblet cells in stained sections was considerable variable between individual organoids, posing challenges for quantification. Quantification of FCGBP showed an increased protein abundance as revealed by larger FCGBP-positive area while showing similar intensity indicating a potentially higher FCGBP content in the granules (Figs. 4C and EV3D). Analysis of MUC2 staining showed a significant decrease in intensity indicating an alteration in secreted mucus or changes in goblet cell function (Fig. 4D,E). The quantity of enteroendocrine cells, another type of secretory cell, decreased significantly, as did the amount of CHGA per enteroendocrine cell. (Fig. 4F–H). Overall, these results indicated that SLE serum stimulation alters the differentiation dynamics towards the secretory lineage and suggested a change in mucus composition marked by an increase in FCGBP and a reduction in MUC2 as well as other factors secreted to it.

## Secretory lineage differentiation is affected 24 h after stimulation

To better understand the underlying mechanism, we stimulated organoids for 24 h with SLE or control serum. This gave us the opportunity to confirm that we were facing an effect on the differentiation process marked by altered transcription factor expression rather than cell-type loss through increased cell death. Sequencing analysis indicated that even brief stimulation could trigger a distinct response, evidenced by the separate clustering in the PCA analysis (Fig. 4I). Among the DEGs we observed an upregulation of genes connected to IFN signaling and several mitochondrial encoded genes indicating their distinct role in initiating the SLE serum changes seen after 72 h (Fig. 4J). Functional analysis showed a stimulation time-dependent decrease of relative spare respiratory capacity similar to that observed after 72 h stimulation (Figs. 4K and 3D). In addition, transcription factors, SPDEF, ATOH1, and HES1, all important for the induction of differentiation, were downregulated, indicating a delayed or altered differentiation towards specialized cells (Fig. 4J).

In summary, these results support the hypothesis that SLE serum stimulation-induced alterations in differentiation, particularly impacting the secretory lineage. Short-term serum stimulation induced changes in transcription factor expression necessary for differentiation which upon long-term stimulation led to alterations in cell-type composition of the organoids. Ultimately, these changes will accumulate and result in an altered mucus layer that in vivo has the potential to trigger alterations in the protective function of the mucus and influence the microbiome.

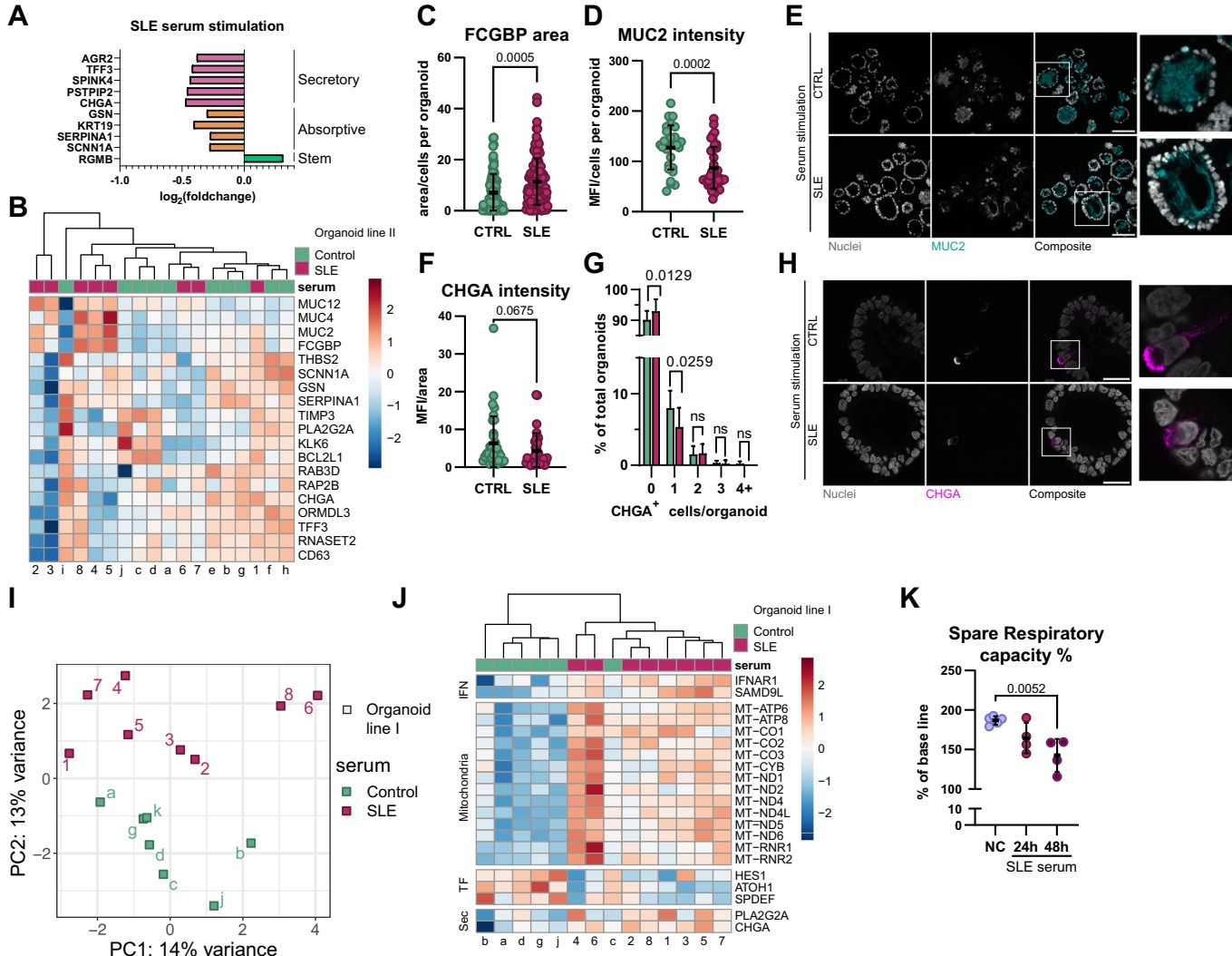

**Figure 4. SLE serum stimulation impacts differentiation dynamics, particularly affecting the secretory lineage.**

(A) Bar graph showing expression changes of selected differentially expressed cell-type markers comparing SLE to control serum-stimulated organoids. Pooled results of organoid line I and II are shown. (B) Heatmap with hierarchical clustering of downregulated genes associated with the gene set "Secretory Vesicle" and "Mmucus" comparing SLE to control serum-stimulated organoids from organoid line II. (C) Quantification of FCGBP-covered area normalized to nuclei per organoid. Each dot corresponds to a single organoid (organoid line II) stimulated with control (n = 76) or SLE serum (n = 129). Data are represented as mean ± SD, and statistical significance determined by Kolmogorov–Smirnov test. (D) Quantification of MUC2 mean intensity per organoid normalized to the number of nuclei per organoid. Each dot corresponds to a single organoid (organoid line II) stimulated with control (n = 29) or SLE serum (n = 40). Data are represented as mean ± SD, and statistical significance determined by unpaired t test. (E) Immunofluorescence of MUC2 in either control (top) or SLE (bottom) serum-stimulated organoids. Nuclei marked by Hoechst33342 in gray and mucus by MUC2 in cyan. Insets showcase enlarged areas, scale bars = 100 μm. (F) Quantification of CHGA intensity normalized by the area of the CHGA granule. Each dot corresponds to a single organoid (organoid line II) stimulated with control (n = 36) or SLE serum (n = 38). Data are represented as mean ± SD, and statistical significance determined by unpaired t test. (G) Distribution of the number of CHGA positive cells per organoid normalized to total organoid number. Organoids stimulated with SLE (n = 801) or control serum (n = 1270) were compared. Data are represented as mean ± SD, and statistical significance determined by two-way ANOVA with Šídák's multiple comparisons test. (H) Immunofluorescence of CHGA in either control (top) or SLE (bottom) serum-stimulated organoids. Nuclei marked by Hoechst33342 in gray and CHGA in magenta. Insets showcase enlarged areas, scale bars = 25 μm. (I) PCA analysis performed on organoids stimulated for 24 h with control (green) and SLE (pink) serum (organoid line I). (J) Heatmap with hierarchical clustering of genes connected to IFN signaling, mitochondria, transcription factors related to differentiation (TF) and secretory lineage (sec) comparing SLE to control serum-stimulated organoids from organoid line I. (K) Seahorse assay showing relative spare respiratory capacity comparing unstimulated (NC, gray) to organoids stimulated with SLE serum for 24 h (light pink) or 48 h (dark pink). Data are represented as mean ± SD, and statistical significance determined by ordinary one-way ANOVA with Tukey's multiple comparisons test. Source data are available online for this figure.

## Effects on epithelial cells depend on IFNAR1, although not being exclusively linked to IFN-α activity

The importance of type I IFN in SLE pathogenesis and its potential as target for treatment was highlighted by the approval of

anifrolumab, a type 1 interferon receptor (IFNAR1) antagonist in 2021 (Burki, 2021). Our novel disease model was able to show the involvement of type I IFN in altering epithelial cell signatures, as suggested by the increased IFNSG expression and activation of the IFN signaling cascade (Fig. 2A–E). To verify

this, we utilized the inhibitory potential of anifrolumab on IFN signaling. We validated that anifrolumab was able to inhibit phosphorylation of STAT1, a downstream target of IFNAR1 activation (Fig. EV3E) by organoid stimulation with up to 100 ng/ml IFN-α2b. Analysis of the expression profile showed no significant difference between organoids stimulated with SLE or control serum when additionally treated with anifrolumab (Fig. 5A). To further validate the successful inhibition of IFNAR1 signaling we checked normalized counts of DEGs specific for epithelial IFN signaling identified before (Figs. 5B and 2B). We could see that anifrolumab was able to abolish the increased expression of these genes not only in IFN-stimulated organoids but more importantly also in combination with SLE serum stimulation making them indistinguishable to the control condition (Fig. 5B). As a next step we wanted to assess if anifrolumab was also able to revert the mitochondrial dysfunction. The respiratory assay confirmed that with blocking IFNAR1 SLE serum-stimulated organoids had a similar relative spare respiratory capacity as in the control condition (Fig. 5C). In addition, to confirm that the effect we were looking at was highly specific to SLE, we analyzed the effect of serum from patients suffering from granulomatosis with polyangiitis (GPA), a systemic autoimmune disease not driven by type I interferon. We could show that organoids stimulated with serum from GPA patients showed no distinctive expression pattern (3 DEGs with $P_{adj} < 0.05$) and no mitochondrial dysfunction (Fig. EV3F).

To confirm that the observed effect was specific to SLE serum, rather than simply a consequence of IFN independently, we stimulated organoids with IFN-α2 in combination with control serum (IFN-α+serum). However, this stimulus was not able to induce the same complex response as seen for SLE serum. Overlapping the DEGs from both comparisons, SLE compared to control serum and IFN-α+serum compared to IFN-α+serum +anifrolumab, showed only a small overlap of almost exclusively IFN-related genes highlighting the unique response of the organoids to SLE serum (Fig. EV3G).

Collectively, these findings suggest that the response of the organoids to SLE serum is dependent on IFNAR1, but not exclusively results from its activation by IFN-α. This implies that the unique composition present in the SLE serum plays a significant role in provoking the response, in conjunction with the dependency on IFNAR1.

## Colon organoid scRNA-seq reveals the presence of all major colonic cell types found in vivo

To examine the cellular composition of the organoids, as well as to gain a deeper understanding of the cell types that could potentially exhibit higher sensitivity to serum stimulation, we conducted scRNA-seq. This approach allowed us to analyze expression patterns of individual cells and gain insights into their specific responses to serum stimulation. We could assign nine clusters to the cells analyzed, which showed a distinct cell cycle distribution and expression profile specific to their cell type (Figs. 6A–C and EV4A,B). Trajectory analyses identified lineage transitions and showed connectivity between progenitor and differentiated populations (Figs. 6D and EV4C). As anticipated, stem cells gave rise to transit-amplifying cells (TA1, TA2, and TA3), the secretory lineage (GC), and the absorptive lineage comprising three colonocyte clusters (eCL, cCL, and ncCL).

We identified two distinct clusters of stem cells (SC1 and SC2) marked by the expression of classical stem cell markers LGR5, OLFM4, RGMB, and SMOC2 (Fig. 6A,B). The key characteristic that sets apart these two clusters is their distinctive cell cycle distribution. While a significant portion of cells in SC1 are in G1 phase, the majority of cells in SC2 are in S phase (Fig. 6C). Transition to cycling TA cells (TA1) was marked by the transition to exclusively G2/M phase and an enrichment of the proliferation markers MKI67 and TOP2A expression (Fig. 6B,C). Furthermore, we identified two additional TA clusters, TA2 and TA3 (Fig. 6A). All three TA clusters showed gene expression gradients along with distinct cell cycle distributions reflecting active proliferation and

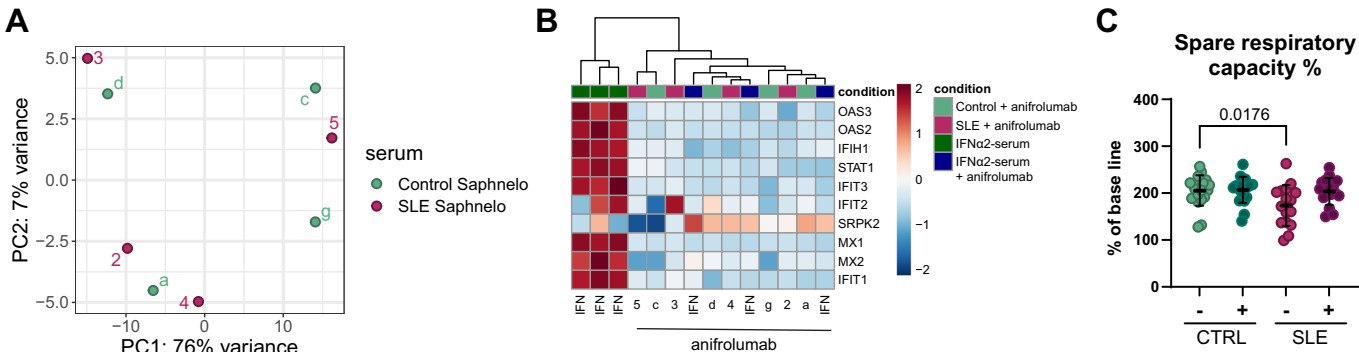

**Figure 5. Impact of anifrolumab on blocking IFNAR1 in organoids stimulated with either SLE or control serum.**

(A) PCA analysis performed on organoids treated with anifrolumab (IFNAR1-inhibitor) in addition to serum stimulation for 48 h in organoid line II. (B) Heatmap with hierarchical clustering of IFN-signature gene expression showing IFN-α2, control or SLE serum-stimulated organoids from organoid line II treated with anifrolumab as indicated. (C) Seahorse assay showing relative spare respiratory capacity comparing control (green) and SLE (pink) serum-stimulated organoids with and without anifrolumab treatment. Data are represented as mean ± SD, and statistical significance determined by one-way ANOVA with Holm–Šidák's multiple comparisons test. Each dot corresponds to one well of organoids stimulated with control ($n = 10$) or SLE serum ($n = 8$). Each serum stimulation was analyzed with $n = 2$ technical replicates. (A, B) $n = 3$ for control serum+IFN-α2 stimulation with or without anifrolumab, $n = 4$ for control (a, c, d, and g) and $n = 4$ for SLE (2-5) serum with anifrolumab. Source data are available online for this figure.

transition to progenitors of absorptive and secretory lineage (Figs. 6B,C and EV4C). Cell transition to G1 phase marked all identified differentiated cell types, early colonocytes (eCL), canonical colonocytes (cCL), noncanonical colonocytes (ncCL), and goblet cells (GC) (Fig. 6C).

Interestingly, we could identify two different types of colonocytes emerging from eCLs (Fig. 6D). Canonical colonocytes were marked by the expression of classical absorptive markers SLC26A3, CA2, and FABP1. These markers were absent in the ncCLs which were marked by high expression of MMP7, LYZ, IL32, and the IFN response genes ISG15, ISG20, and IFI27 (Fig. 6B). Even though MMP7 and LYZ were lately described to be present in deep crypt secretory cells (Sasaki et al, 2016; Schwank et al, 2013), our cell population lacked other reported markers of these newly described cells. In addition, we could observe the expression of HES1, an exclusive marker of the absorptive lineage (Fig. 6B), suggesting that we were looking at a different cell type.

Further analysis of the significantly enriched markers ($P_{adj}$<0.1 and log$_2$fc > 0.25) suggested different functions of the two cell types. Canonical colonocytes showed an enrichment of lipid metabolism-related pathways and expressed several ion transporters. Specifically, SLC26A3 and CA2 which in vivo are expressed by fully mature colonocytes, and are central players of absorption (Van Der Post et al, 2019), suggesting that in vitro their function would also focus on lipid processing and absorption. In contrast, we could observe an enrichment of pathways connected to immune defense in ncCL indicating cytokine-driven responses to maintain barrier integrity (Fig. 6E).

The secretory lineage was characterized by the presence of classical goblet cell markers MUC2, SPINK4, SPDEF and ATOH1. In addition, we could see the expression of ZG16, FCGBP, and CLCA1, all mucus components (Fig. 6B). Further subclustering of the goblet cell population resulted in five clusters (Figs. 6F and EV4D). These could be assigned to two early goblet cell types (GC0 and GC3) and two types of differentiated goblet cell populations (GC1 and GC2) (Figs. 6G and EV4E). Analyzing their gene expression pattern revealed that cluster GC1 had increased expression of AGR2, FCGBP, CLCA1, ERN2, and SPDEF (Fig. 6G) all known to be mucus core proteins or genes involved in mucus biosynthesis (Gustafsson and Johansson, 2022). In contrast to that cluster GC2 was characterized by a higher expression of ZG16 and TFF3 which function as AMP and regulator of mucus viscosity, respectively (Kurashima and Kiyono, 2017; Bergström et al, 2016). Cluster GC4 showed only low expression levels of typical goblet cell markers. With further analysis we were able to identify a small fraction of cells within this cluster expressing CHGA and POU2F3 indicating the presence of enteroendocrine and tuft cells, respectively (Fig. EV4F). Interestingly, similar as reported previously for human colon tissue (Nyström et al, 2021; Burclaff et al, 2022; Birchenough et al, 2015) we were able to see differences in mucin gene expression patterns between the goblet cell cluster, indicating their distinct functionality also in vitro (Fig. 6H).

In summary, we identified nine unique clusters composed of stem cells, TA cells, goblet cells, canonical, and noncanonical colonocytes. In addition, we observed five subclusters within the secretory cell population including enteroendocrine and tuft cells as well as different goblet cell subtypes. Thus, our in vitro model represents the majority of the epithelial cell types found in the descending colon providing a valuable model to study the effect of SLE serum on an in vivo like intestinal epithelial barrier.

## scRNA-seq analysis unveils diverse cellular responses leading to an alteration in mucus composition

After confirming the presence of the major colonic cell types in our near-physiological in vitro model we wanted to understand which impact the SLE-specific serum signature would have on different epithelial cell types. Differential gene expression analysis within all identified cells revealed 480 genes (with a $P_{adj} < 0.1$ and average log$_2$fold change >|0.1|) that were misregulated upon SLE serum stimulation. We observed downregulation of pathways related to protein translation and altered expression of genes connected to oxidative phosphorylation specifically in stem cells, early colonocytes and goblet cells (Fig. EV5A,B). Both protein translation and mitochondrial function are important players in modulating proliferation and differentiation underlining the altered cell-type composition upon SLE serum stimulation (Blanco et al, 2016; Sünderhauf et al, 2021). Further analysis showed that some genes were broadly altered like MIF, Macrophage Migration Inhibitory Factor, the tight junction protein CLDN3 or AQP5 which were significantly downregulated (Fig. 7A). However, we could see that each cell type had a specific response to SLE serum stimulation. In SC1, but not in SC2, stem cell markers SMOC2 and OLFM4 were significantly downregulated. In contrast, in cCL typical absorptive markers were unaltered while tight junction proteins TJP1 and CLDN7 were downregulated. Both are important for barrier integrity and their absence is associated with a delay in mucosal repair or an initiation of inflammation in vivo (Tanaka et al, 2015; Kuo et al, 2022). This specifically suggests an alteration in the epithelial barrier with an increase in leakiness and lays the ground for a self-perpetuating chronic inflammation as seen in SLE.

Next, we wanted to understand if we could detect a change in goblet cell differentiation as indicated by the bulk RNA sequencing data. We observed a noticeable decline in the levels of goblet cell markers such as TFF3, ZG16, MUC2, and FCGBP within the TA population (TA1, TA2, and TA3) (Fig. 7B). This suggested that the stimulation with SLE serum led to diminished differentiation towards the secretory lineage, which in turn suggested a potential reduction of cells differentiating towards goblet cells. Our findings were in line with the results from the bulk RNA sequencing analysis, wherein we noticed a reduction in the expression of early goblet cell markers during the initial phase of stimulation (24 h), and a subsequent decrease in late secretory markers with extended stimulation (72 h) (Fig. 4A,B,I). Consistent with these results, we could detect a minor reduction in the total number of cells within the goblet cell cluster (Figs. 7C and EV4A). Interestingly, the diminished goblet cell population showed a significant upregulation of FCGBP, an IgGFc-binding protein, which is crucial for structural integrity of the mucus layer (Liu et al, 2022; Ehrencrona et al, 2021). In addition, WFDC2 which was only recently identified to be required for barrier integrity and is downregulated in goblet cells of UC patients (Parikh et al, 2019) was also downregulated upon SLE serum stimulation (Fig. 7D). Further analysis showed an overall trend towards a cell population decrease across all goblet cell subclusters, with a significant reduction in GC4 (Fig. EV5C). GC4 lacked the expression of classical goblet cell markers and contained CHGA-expressing cells, suggesting a reduction in enteroendocrine

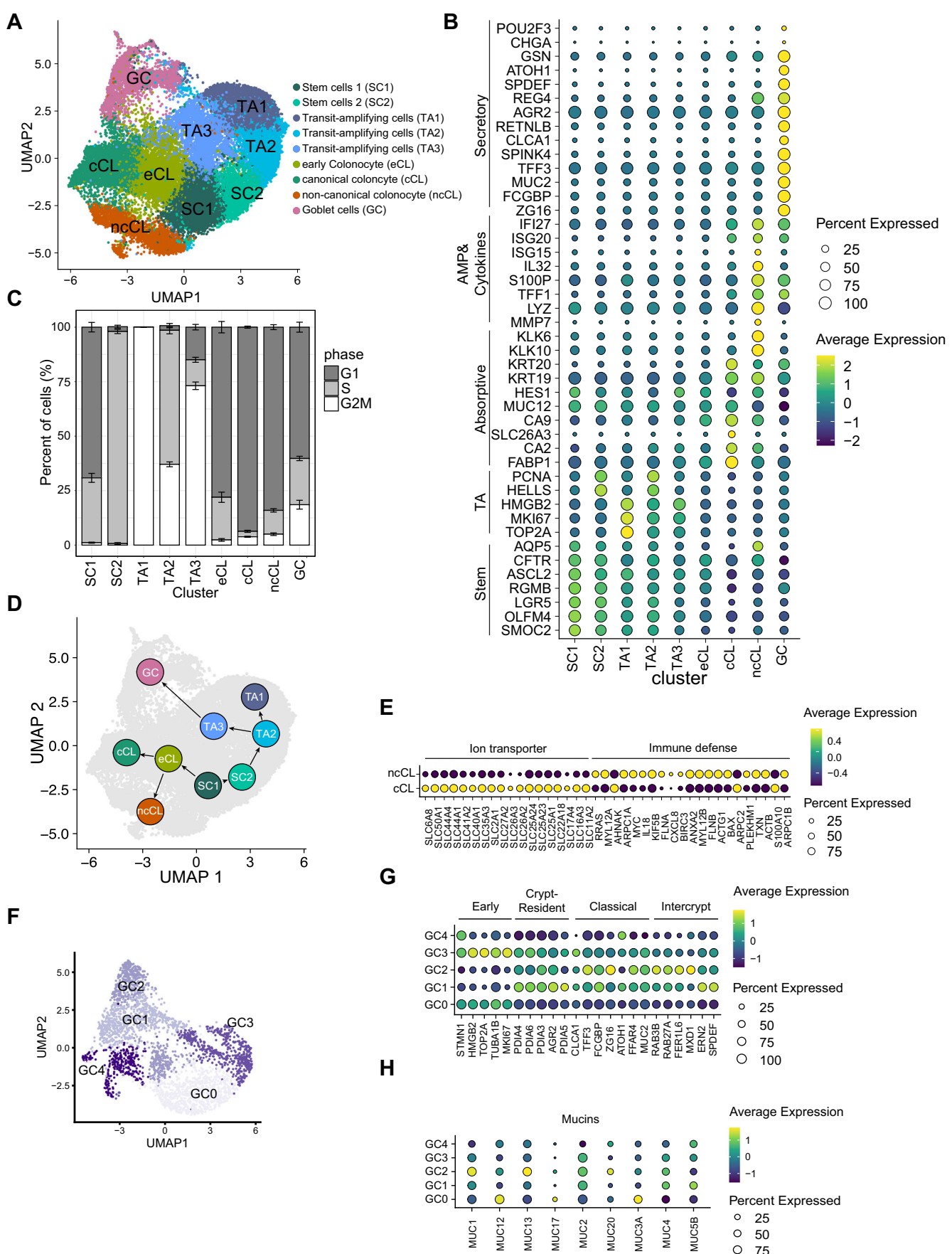

◄ **Figure 6. scRNA-seq reveals diverse cellular composition in colon-derived organoids.**

(**A**) UMAP analysis performed on all sequenced epithelial cells derived from organoids stimulated with SLE ($n = 3$) or control ($n = 3$) serum from organoid line II. Identified clusters were associated to cell types as shown. (**B**) Dot plot illustrating the predominant marker genes corresponding to each cluster in colon-derived organoids. (**C**) Cell cycle phase distribution for each cell-type cluster. Data are shown as mean ± SEM, $n = 6$. (**D**) UMAP plot illustrating the different predicted cellular developmental trajectory using Slingshot. (**E**) Dot plot of selected marker genes that distinguish canonical and noncanonical colonocytes. (**F**) UMAP analysis showing subclustering of goblet cells of all analyzed samples. (**G**) Dot plot representing the average expression of goblet cell subtype marker genes across identified goblet cell subclusters. (**H**) Dot plot displaying the average expression levels of mucin genes across goblet cell subclusters. The size of the dots in (**B, E, G, H**) correspond to the proportion of cells expressing a specific gene, while the color indicates the mean relative gene expression across the respective cell type. All plots show the data from all analyzed cells independent of the serum source.

cells as identified by staining (Fig. 4F). In addition, GC4 showed no DEGs while the DEGs of the other GC clusters were mainly downregulated ribosomal genes connected to translation and protein translocation. This might indicate an impact on goblet cell function that highly depends on constant translation and protein trafficking. Apart from the reduced number of goblet and enteroendocrine cells, these findings suggest alterations in the mucus layer.

Since mucus properties can easily be modified by an altered abundance of mucins and AMPs (Leal et al, 2017; Meyer-Hoffert et al, 2008) which is not restricted to goblet cells we checked known markers in the absorptive lineage. In the early colonocyte population, we observed a downregulation of MUC12 a membrane-bound mucin which is part of the protective glycocalyx (Dotti et al, 2017). In addition, we observed an increase of lysozyme (LYZ) and decrease in REG4 in the cCL (Fig. 7D). Both are known to be secreted into mucus and are important for microbial defense (Ferraboschi et al, 2021; Wang et al, 2022a). A similar alteration was also evident within the ncCL where we observed an alteration in the expression of MMP7, RETNLB and PLA2G2A (Fig. 7D), genes that are essential components of barrier function, especially RETNLB and PLA2G2A with their bactericidal properties (Mahapatro et al, 2016; Vandenbroucke et al, 2014; Xiao et al, 2022; Propheter et al, 2017). In addition, in ncCL and GC2 we were able to detect the induction of IFI27, an interferon-inducible gene, (Fig. 7D) similar as reported for whole blood samples from SLE patients (Becker et al, 2013; Ishii et al, 2005; Wang et al, 2022c) and as observed by the induction of IFNSG in our bulk RNA sequencing data (Fig. 2C). Interestingly, ncCL and GC2 both expressed AMPs which are known to be altered by pro-inflammatory cytokines (Muniz et al, 2012; Kolls et al, 2008; Hu et al, 2021) suggesting a similar connection in our model.

Overall, these results revealed that SLE serum stimulation of epithelial cells lead to distinct transcriptional alterations which are marked by a shift in differentiation as already suggested by our bulk RNA sequencing data. The observed expression changes in goblet cell markers, mucins and AMPs in both, the secretory and absorptive lineage, along with the mild reduction of goblet cell number might integrate in an ultimately changed mucus composition and compromise its function as first line of defense. In vivo such changes could have detrimental effects on the barrier integrity, immune response, and the gut microbiome.

### The intestinal epithelial barrier is altered in descending colon tissue biopsies from SLE patients

To highlight the relevance of this disease model and provide context regarding intestinal involvement in SLE, we conducted an expression analysis of intestinal biopsies derived from descending colon of healthy controls ($n = 5$) and SLE patients ($n = 3$). We could identify 256 DEGs which represented the integrated response of all present cell types in biopsies. In our further analysis we focused on genes that were expressed by epithelial cells. Overall, we could observe downregulation of several genes connected to absorption and ion transport (SLC1A1, SLC13A2, SLC16A9, SLC25A20, SLC25A34). BEST2, a marker for goblet cells, was significantly downregulated while markers for enteroendocrine cells (SGC5, GCG, VWA5B2) were upregulated, suggesting an alteration in the secretory lineage (Fig. 7E). In addition, we observed reduced expression of ion transporters SLC26A3, SCNN1A, and SCNN1B while other markers for colonocytes were unaltered. This indicated an alteration in cell function rather than a loss of colonocytes. Those three ion transporters are the major players in water absorption in descending colon and are able to modulate mucus viscosity, ion content and ultimately alter its structural composition via changes in sodium absorption and bicarbonate secretion influencing the mucosal pH (Van Der Post et al, 2019; Zhou et al, 2011). In addition to that several factors important in the maintenance of the intestinal stem cell niche and differentiation (WNT5A, RSPO2, GREM1, BMP5) were significantly downregulated suggesting changes in the physiological proliferation and differentiation dynamics. We then checked the expression of the top secretory cell markers identified with single-cell sequencing. Surprisingly, we could see clustering of two SLE patient samples that showed a reduction in the selected markers (Fig. 7F).

The evidence we have gathered collectively suggests that the impact of SLE is not confined to immune cells, but also affects intestinal epithelial cells. The combined findings of our in vitro model and the analysis of the patient biopsies indicate that SLE pathogenesis involves epithelial cells and could potentially weaken the structural integrity of the mucus layer and disrupt the barrier function. The increased barrier permeability seen in vitro reflects the reported alteration in intestinal barrier function in SLE patients. Taken together, our results shed light on the broader implications of SLE, revealing that its effect extends beyond the immune cell compartment affecting the intestinal epithelial barrier.

## Discussion

The vital function of the intestinal epithelial cells is to form a barrier that enables tightly regulated passage of a variety of molecules. The barrier integrity highly depends on the presence of a mucus bilayer consisting of mucin, mucus-associated proteins and antimicrobial peptides (Gorman et al, 2023). Lack of the core mucus protein MUC2 leads to the spontaneous development of

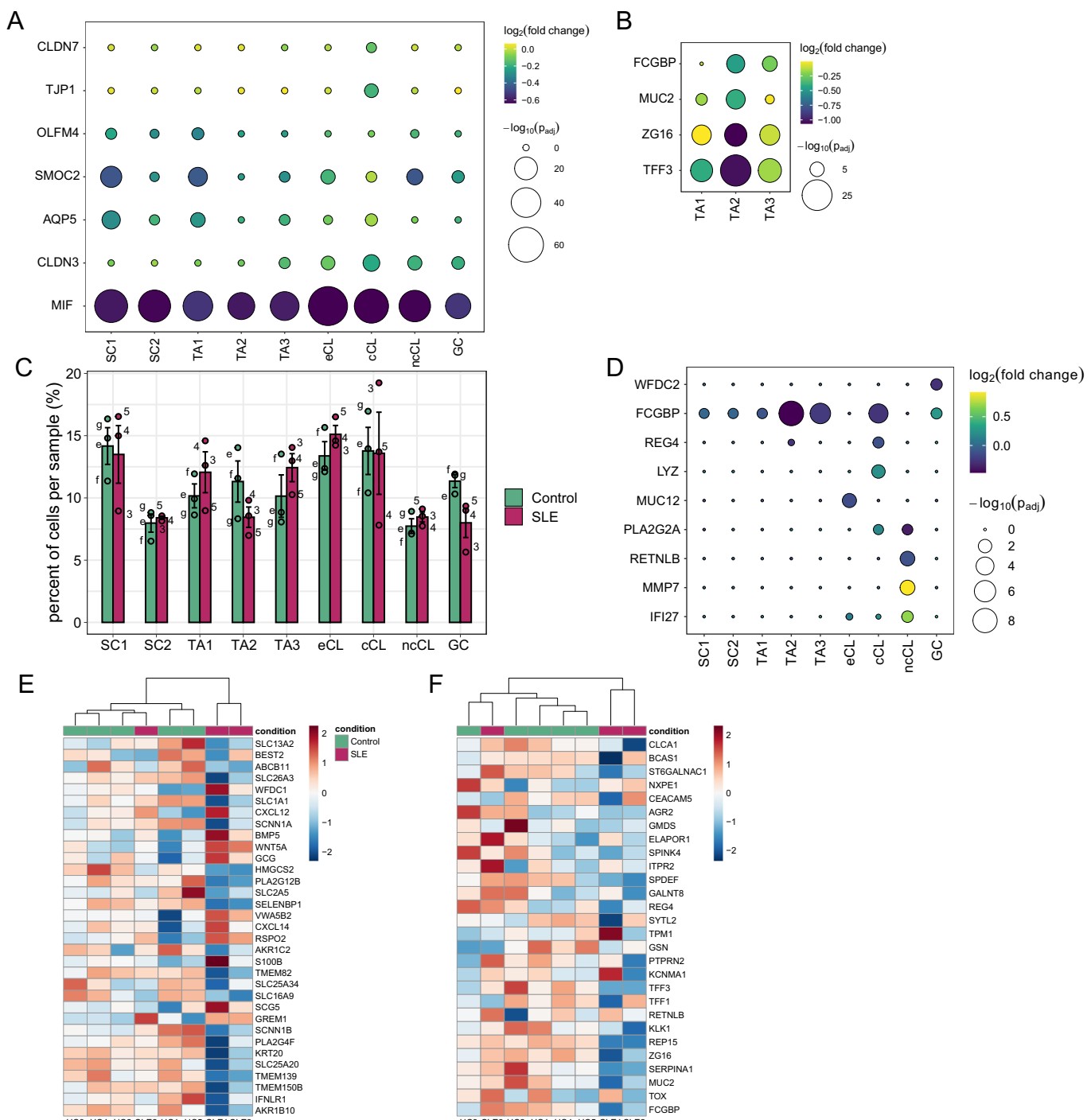

**Figure 7. Impact of SLE serum stimulation on cellular composition and goblet cell diversity.**

(A) Dot plot showing DEGs for each cell type comparing SLE to control serum-stimulated organoids. (B) Dot plot showing goblet cell-specific DEGs comparing SLE to control serum-stimulated organoids. (C) Analysis showing cell-type ratios comparing SLE ($n = 3$; serum samples 3–5) to control ($n = 3$; serum samples e–g) serum.-stimulated organoids. Data are presented as mean with ±SD. (D) Dot plot showing mucus-related DEGs comparing SLE to control serum-stimulated organoids. (E) Heatmap with hierarchical clustering of selected DEGs comparing SLE ($n = 3$) to control ($n = 5$) descending colon biopsies. (F) Heatmap with hierarchical clustering of genes independent of their significant differential expression levels. Genes are associated with goblet cells and were selected based on the findings in vitro. (A, B, D) The size and color of each dot represent $P_{adj}$ value and fold change, respectively.

inflammation in the colon, and alterations in the mucus layer are associated with a dysfunctional intestinal barrier (Van der Sluis et al, 2006; Van Der Post et al, 2019; Singh et al, 2022; Pullan et al, 1994; Lo Conte et al, 2023). Recent reports show an increased intestinal permeability in SLE patients (Azzouz et al, 2019; Shi et al, 2014; Issara-Amphorn et al, 2018; Silverman et al, 2019; Nockher et al, 2008). However, not much is known about the cause and consequence of this observation. Systemic inflammation is marked by hyperactivity of the immune system which is reflected in an altered serum cytokine profile. We hypothesize that the constant presence of pro-inflammatory cytokines in the bloodstream and influx to the intestine not only impacts the immune cells in the lamina propria, but it also prompts changes in the epithelial cells. We implemented a model that stimulates organoids with serum to explore primary effects of serum-contained factors on the constantly regenerating epithelial barrier.

With our single-cell sequencing analysis, we could observe the presence of nine distinct cell populations. Among them were the main cell types of the colon, intestinal stem cells, TA cells as well as cells of the secretory and absorptive lineage. Within the goblet cell population, we could identify not only different goblet cell types with specific function but also a small subfraction of enteroendocrine and Tuft cells. The absorptive lineage was composed of three distinct population whereof one could be assigned to colonocyte progenitors (eCL) and the other two as differentiated colonocytes. Among these two distinct clusters, we identified a canonical colonocyte population that was highlighted by the expression of markers such as SLC26A3, CA2, and FABP1. Interestingly, the second colonocyte population was marked by a unique expression profile with increased expression of REG4, MMP7, LYZ, and the cytokines IL-18, IL-23, IL33 (Dataset EV7). For these genes it has been shown that they can be induced by pro-inflammatory cytokines in vitro and in vivo (Schumacher et al, 2023; Takasawa et al, 2022; Jin et al, 2020; Dotti et al, 2017; Mahapatro et al, 2016; Hasegawa et al, 2011; Xiao et al, 2022). In addition, we could observe the presence of interferon-inducible genes IFI27, ISG15, and ISG20. These findings indicate that the expression profile of this noncanonical colonocyte population was shaped by the cytokine cocktail contained in the serum and identifies a cell population with a specialized function connected to immune defense. Deeper analysis is needed to classify the relevance and function of this cell type in vivo. Further studies will be required to fully understand the implications of these findings and their possible relevance in the generation of complex in vitro models. Nevertheless, given that most cell types of the descending colon were represented in our model, we could utilize it to study the effects of SLE serum on the intestinal epithelial barrier.

SLE is a multifaceted autoimmune disease which central player is IFN-α (Oke et al, 2019; Crow, 2014; Reynolds et al, 2018). The induction of IFNSG in immune cells has been used as a reliable tool to identify increased serum IFN-α levels as it is the case for the majority of the patients (Buang et al, 2021). Our study shows that serum of SLE patients is able to induce a similar IFNSG in epithelial cells in vitro. Thus, intestinal organoids represent a novel tool to assess type I IFN levels in patient serum and to improve our understanding of the reported interindividual heterogeneity in the response to IFN-α (Jarry et al, 2017). In addition, we could show that changes induced on the epithelial cells are dependent on type I IFN signaling and can be inhibited by the treatment with anifrolumab an inhibitor of IFNAR1. We further confirmed type I interferon dependency by the absence of DEGs after the stimulation of organoids with serum derived from Granulomatosis with Polyangiitis patients. However, IFN-α in combination with control serum was not able to reproduce the phenotype induced by SLE serum indicating a synergistic effect of type I IFN with other cytokines contained in SLE serum. IL-6, IL-10, IL-18, and IL-23 levels were elevated, and their receptors expressed by the organoids, hence the observed response reflects most likely the integration of all their effects. It was intriguing to see that cytokines with a concentration <3 pg/ml were able to induce such a pronounced phenotype demonstrating their potency. In addition, this underlines the artificial environment created using high levels of a cytokine or even a cytokine cocktail for stimulation questioning their relevance in modeling pathological conditions in vitro. While they have their relevance in understanding underlying signaling pathways, novel near-physiologic in vitro models like the one developed in this study will be of high importance in the future to mimic complex diseases. Our model provides intriguing avenues for exploration into the pathogenesis of SLE.

So far, the focus of SLE research has been the dysregulation of the immune system, but studies in mouse intestine and lung cells show that epithelial cells can be affected as well by type I IFN leading to detrimental changes in the barrier (Katlinskaya et al, 2016b; Major et al, 2020). We could show here that this holds also true for human intestinal cells. However, effects seen upon SLE serum stimulation were not exclusively resulting from IFN-α but were specifically caused by the unique composition of serum as mediator for the systemic inflammation. We observed that mitochondrial pathways and function in the intestinal organoids were affected. SLE serum stimulation caused an increased metabolic activity marked by higher basal respiration and reduced relative spare respiratory capacity highlighting the capacity of serum to induce metabolic changes. A similar mitochondrial dysfunction was recently also reported in CD8 + T cells derived from SLE patients (Buang et al, 2021). In addition, our data suggested that the alteration induced by SLE serum was different from the metabolic shift induced by differentiation. Similarly as reported for the transition from proliferation to differentiation in vivo, we could see that differentiation led to a shift from glycolysis to oxidative phosphorylation (Sünderhauf et al, 2021; Rath et al, 2018; Rodríguez-Colman et al, 2017). In contrast SLE serum stimulation induced only a slight increase in oxidative phosphorylation but was more dominated by an increase in glycolysis. In UC patients a similar shift towards increased glycolytic activity is accompanied by an impaired goblet cell differentiation (Sünderhauf et al, 2021) suggesting that the seen changes might be caused by the altered cell-type composition of the SLE serum-stimulated organoids. Mitochondrial analysis at single-cell resolution would be required to explore if this shift is caused by an overall alteration of mitochondrial function.

We were intrigued to see that the most common alteration, which was persistent in all our experiments, was the involvement of the secretory cell lineage. The downregulation of several markers connected to secretion, goblet and enteroendocrine cells identified with bulk RNA sequencing aligned with our results from microscopy and scRNA-seq which revealed a diminished goblet and enteroendocrine cell population and a decrease or delay in the differentiation towards secretory cells. We could show that a

subcluster of GCs and ncCLs were specifically responsive to IFN contained in the serum by the expression of IFI27, indicating a cell-type-specific response. In addition, our results indicated a functional change of differentiated cells resulting in an altered mucus layer. Those changes were not restricted to the goblet cell population but extended to the absorptive cell lineage. We classified numerous misregulated genes, many of which were integral to mucus composition (AGR2, SPINK4, MUC12, FCGBP) and antimicrobial defense (PLA2G2A, RETNLB, REG4, TFF3, and WFDC2), suggesting a functionally changed mucus layer and a potentially altered microbial niche. Our data aligned with the altered expression pattern of several mucus-related genes and the reported metaplastic, colonic expression of LYZ associated with UC pathogenesis (Van Der Post et al, 2019; Dotti et al, 2017; Kurashima and Kiyono, 2017; Sarvestani et al, 2021; Arnauts et al, 2020; Ojo et al, 2021). We observed downregulation of MUC12, a membrane-bound mucin that has a protective role for colonocytes (Yamamoto-Furusho et al, 2015) and an upregulation of FCGBP which is important for mucus integrity (Dotti et al, 2017; Sarvestani et al, 2021; Arnauts et al, 2020; Kim et al, 2006). MMP7 which is associated with LPS-induced barrier permeability and connected to SLE pathogenesis (Vandenbroucke et al, 2014; Vira et al, 2017) itself was upregulated. The additionally identified downregulation of the sodium channel SCNN1A and aquaporin AQP5 suggests an alteration in ion composition and hydration of the mucus (Baker et al, 2012; Pluta et al, 2012). The changes were not restricted to the mucus composition but extended to the epithelial barrier itself. With scRNA-seq we were able to identify downregulation of tight junction proteins CLDN3, CLDN7, and TJP1 which could be the cause for the increased leakiness observed upon SLE serum stimulation. Using FITC in our permeability assay enabled us to evaluate not only the leak pathway but also allowed us to investigate the pore pathway which depends on claudins and regulates the passage of solutes up to 0.6 nm (Pongkorpsakol et al, 2020). This was of specific interest since a recent report using intestine-specific Cldn-7 knockout mice showed increased intestinal permeability which was only detectable by FITC (389 Da) and not by the commonly used bigger FITC-dextran 4 kDa (Tanaka et al, 2015). Interestingly, in this model, they reported an increase in bacterial chemotactic peptide influx which was connected to the observed intestinal inflammation showing the potential connection between gut leakiness and inflammation.

These findings show that SLE serum stimulation led to an alteration in both the secretory and absorptive lineage that resulted in a potential structural weaking of the mucus barrier. Not only the differentiation to goblet cells and therewith a reduction in their number was affected but also several secreted factors necessary for a functional mucus layer. The alteration in expression of AMPs suggested changes in intestinal antimicrobial capacity of the mucus which is pivotal to maintain the sterility of the inner mucus layer and therewith reducing the microbial-epithelial interaction in vivo (Johansson et al, 2008). It is important to mention that in UC it is suggested that mucus alterations are causative for disease pathogenesis and precede the active disease, indicating the importance of the mucus layer for barrier function and intestinal homeostasis (Van Der Post et al, 2019). Also other diseases like cystic fibrosis highlight the importance of a physiological mucus composition in order to maintain organ function and underline that even minor changes to the mucus layer can have potentially

detrimental effects on barrier function (Fernández-Blanco et al, 2018). Additional studies are required to confirm changes in the physical properties of the mucus layer in our model system and to understand how these changes might impact the epithelial barrier in vivo. It would be of specific interest to unravel if changes in the mucus layer could trigger changes in microbiome composition reported for SLE (Pan et al, 2021; Tomofuji et al, 2021; Wang et al, 2022b). The observed increased epithelial permeability with our in vitro model reflects reports of gut leakiness in SLE patients (Azzouz et al, 2019; Shi et al, 2014; Issara-Amphorn et al, 2018; Silverman et al, 2019; Nockher et al, 2008). However, more research is required to explore which role it plays and if this increased permeability is able to induce an enhanced flux of bacterial products. Taken together, our results suggest a multilayered alteration affecting the mucus layer and the integrity of the epithelial barrier. In vivo such dysfunction could lead to dysbiosis and immune cell response potentially generating an inflammatory environment.

The relevance of the changes unraveled with our in vitro model was highlighted by the findings revealed in a small SLE patient population. We observed that in two of three analyzed descending colon samples, we could detect a clear effect on the epithelial cells manifested by misregulation of ion transporters SLC26A3, SCNN1A, and SCNN1B expression along with BEST2, a goblet cell marker, and the reduced expression of several other goblet cell-related markers. It is of high interest to understand if initiated by these changes, the mucus layer composition and function are altered. A bigger and well-stratified patient population with matched controls is necessary to confirm these initial findings that point toward involvement of the intestinal epithelial cells in SLE pathogenesis.

To conclude, our research underscores the immense potential of organoids in translational research, offering a fresh perspective to study diseases that have stumped medical science for decades. The adaptability of organoids permits the incorporation of different factors, such as disease-specific cytokines or even other cell types, further emphasizing their potential in advancing medical research.

## Methods

### Human samples and study approval

Healthy and SLE whole human intestines were obtained from multiorgan donors that were rejected for transplantation due to reasons unrelated to intestinal diseases. Descending colon tissue of a healthy 46-year-old female and a 23-year-old male were used to generate organoid line I and II, respectively. Serum samples were obtained through collaborations with the Division of Rheumatology, Department of Medicine V, University Hospital Heidelberg, Germany (SLE) and Molecular Neuroimmunology Group, Department of Neurology, University of Heidelberg, Germany (controls). Further information about the serum samples can be found in Table EV1.

The protocol for serum collection was approved by the ethics committees of the University of Heidelberg (272/2006 and S-187/2008). Intestines that could not be used for transplantation were procured from human donors via (1) Novabiosis, Inc. (Research Triangle Park, Durham, North Carolina, USA). The acquisition of

the donor intestines was conducted following ethical committee approval from the Organ Procurement Organizations (OPO), in line with the consent and deidentification guidelines established by the OPOs and the United Network for Organ Sharing (UNOS), under the US Transplantation Network framework. Permission for organ donation was granted by the immediate family members of the donors while preserving the patient's privacy. This donation approval aligns with the guidelines provided by the federal organization UNOS and the Federal Drug Administration (FDA). (2) Samples and data from patients included in this study were also provided by the I3PT Biobank and the DTI Foundation (Barcelona, Spain) they were processed following standard operating procedures with the appropriate approval of the Ethics and Scientific Committees.

All research procedures were conducted adhering to the principles stipulated in the WMA Declaration of Helsinki and the Department of Health and Human Services Belmont Report. All participants in the study gave written informed consent before their inclusion and were assured that their identities would remain confidential in relation to this study.

## Organoid generation and culture

Crypt isolation was performed as previously published (Sato et al, 2011). The mucosa of ~1 cm² was separated from the underlying tissue and cut into small pieces, washed with cold PBS before incubation in 10 mM EDTA/PBS for 1 h at 4 °C under constant rocking. The biopsies were then transferred to 5 ml cold PBS, and crypts were liberated by pipetting. This was repeated twice, and the pooled fractions were centrifuged at $250 \times g$ for 5 min at 4 °C. The pellet was once washed in base medium (Table EV2) before the crypts were resuspended in 45 µl 80% Matrigel (#356231, Corning) in expansion medium (Table EV2) and plated in small drops in a 24-well plate. After polymerization, 500 µl expansion medium containing Y-27632 was added until the first passage or later for the first days after passaging. Organoids were cultured at 37 °C and 95% $O_2$, 5% $CO_2$ humid atmosphere. The initial passage was performed when organoids reached a size of ~200–300 µm in diameter. After establishing the organoid line organoids were passaged at a ratio of 1:6-1:10 every 4–5 days by a 45-s incubation in Accutase at 37 °C followed by mechanical dissociation in 6 ml base medium with a 10 ml pipet equipped with a 200-µl pipet tip. Organoid fragments were centrifuged at $150 \times g$ for 5 min at 4 °C and plated as stated above. The medium was changed every 2–3 days. Organoid cultures were tested negative for mycoplasma contamination.

## Organoid stimulation

Organoids were passaged as described above with the modification of excluding bigger-sized fragments by using a 70-µm strainer after the dissociation. In addition, fragments were counted, and 400 fragments/µl were seeded. Organoids were grown in growth medium (Table EV2) for four days before dissociating them into single cells by incubation for 90 s in Accutase followed by mechanical dissociation as described above. Bigger fragments were removed by using a 40-µm strainer. Then 800 cells/µl were seeded in 10 µl Matrigel mix in a 96-well plate. After polymerization, 100 µl expansion medium were added containing Y-27632.

Organoids were stimulated with 5% SLE or 5% control serum in the stimulation medium (Table EV2). In case of IFN-α stimulation, 100 pg/ml IFN-α2B (Humankine, HZ-1072) were added in addition to control serum. Anifrolumab was added at a final concentration of 10 µg/ml. Stimulation started either 24, 48, or 72 h before harvesting organoids on day 5.

## Monolayer generation and permeability assay

Organoids were dissociated into single cells as described above. Then 17000 cells were seeded per 0.11 cm² polycarbonate transwell with 0.4-µm pores (Millipore). Expansion medium was added luminally and basally containing Y-27632, 65 µl, and 200 µl, respectively. On day 3, culture medium was removed and monolayers stimulated basally with 5% SLE or control serum in stimulation medium. Apically stimulation medium without serum was added. The medium was replaced after 48 h. After 24 h, the medium was removed and replaced luminally with 75 µl base medium containing 2 µg/ml fluorescein isothiocyanate (Sigma) and basally with 250 µl base medium. The monolayers were incubated 2 h at 37 °C and 95% $O_2$, 5% $CO_2$ humid atmosphere before the basal reservoir was sampled. The fluorescence of 100 µl medium was analyzed using a plate reader with an excitation filter of 482 nm and emission filter of 520 nm (FLUOstar OPTIMA, BMG Labtech). The mean fluorescence intensity of the controls was then averaged and used to calculate a ratio for all samples.

## Organoid quantification

To analyze organoid morphology, bright-field images of the total Matrigel drop area were acquired at three different z positions in a range of 300 µm with Nikon Eclipse Ti-2 inverted microscope. Images were analyzed using OrgaQuant tool (Kassis et al, 2019) with settings adjusted to 1600 (image size), 2 (contrast), and 0.7 (confidence threshold). The average diameter was calculated.

## LDH cytotoxicity assay

The supernatant of stimulated organoids was collected after 48 h incubation, diluted 1:100 in storage buffer and analyzed as indicated in the manufacturer's protocol (LDH-Glo Cytotoxicity Assay, Promega).

## Immunohistochemistry

Organoids from two wells were pooled in base medium. The organoid pellet was resuspended in 100 µl harvesting solution (Cultrex, Biotechne) and incubated for 45 min at 4 °C. The released organoids were washed with cold PBS, then 4% PFA was added, and organoids were fixed for 45 min at 4 °C followed by a final wash. The organoids were then resuspended in 4% low-melt point agarose. Paraffin embedding was performed following standard protocols. For immunohistochemistry, 3-µm sections were rehydrated, heat-induced antigen retrieval in 10 mM sodium citrate acid and 0.05% Tween 20 (pH 6), and quenching in 50 mM $NH_4Cl$ was performed. Sections were then blocked with 5% goat serum for 1 h. Primary antibodies anti-cleaved Caspase 3 (1:100; Cell Signaling, 9661S), anti-FCGBP (1:200; Atlas antibodies, HPA003564), anti-KI67 (1:200; abcam, ab15580), anti-MUC2 (1:200; Santa Cruz

Biotechnology, sc-515032) or anti-CHGA (1:100; Novus Biologicals, NB120) were incubated overnight at 4 °C. Anti-rabbit-488 (1:500; Invitrogen, A11034) and anti-mouse-568 (1:500; Invitrogen, A-11031) were incubated together with Hoechst33342 (Invitrogen, H3570) for 1 h at room temperature (RT). Mounted sections were imaged using Nikon Ti-2 fluorescence or AX confocal microscope. Image analysis was performed using ImageJ (Bankhead, 2014; Abràmoff et al, 2004). Nuclei were quantified with StarDist (Schmidt et al, 2018) and overlap with KI67 quantified using BioVoxxel tool box (Brocher, 2022).

## Western blot

Organoids were released from the Matrigel as described above. For protein isolation, organoids from four wells per condition were pooled. The organoids were lysed in 120 µl RIPA buffer (Thermo Scientific) containing 1.4× PhosphoStop and cOmplete mini (Roche) for 45 min on ice. Protein concentration was determined using BCA Protein Assay Kit (Pierce). Lysates were diluted in Laemmli buffer, 10 µg were loaded for SDS-PAGE and then transferred to a PVDF membrane. The membrane was blocked with 5% milk in TBST for 1 h at RT and then probed overnight at 4 °C with pStat1 (1:1000; Cell Signaling #9167S), β-actin (1:500; Cell Signaling #4970S), STAT1 (1:1000; Cell Signaling #9167S) or IFIT3 (1:500; Santa Cruz #sc-393512). The signal was detected after incubation with anti-mouse-HRP (1:2000; Cell Signaling, 7076P2) or anti-rabbit-HRP (1:2000; Cell Signaling, 7074P2) for 1 h at RT using ECL solution Clarity Max (Bio-Rad Laboratories) and fluorescence-imaging device (Vilber, FUSION FX7).

## IFN-α/β reporter cell line

IFN-α/β reporter HEK-Blue Cells (InvivoGen, #hkb-ifnab) were selected and grown to confluency in a 96-well plate. Cells were stimulated for 24 h with 10% serum. The supernatant was then mixed with QuantiBlue Solution (Invivogen, #rep-qbs), incubated for 3 h at 37 °C and alkaline phosphatase activity was determined by measuring the absorbance at 600 nm using GloMax®-Multi Detection System (Promega).

## Serum analysis

Serum samples were analyzed using a bead-based multiplex assay (BioLegend, LEGENDPplex, 740809) following the manufacturer's protocol. In total, 5000 events (bead populations A + B) were recorded using a BD FACS Aria Fusion flow cytometer. Cytokine concentration was calculated based on a standard curve using Biolegend's LegendPlex data analysis software. Cytokine levels that were below the detection limit were set to half the detection limit to calculate statistical significance. Cytokine levels for each sample can be found in Table EV3.

## Cytometric analysis

Single-cell suspensions were prepared, and cells stained with MitoTracker green (10 nM) and 50 nM MitoTracker red (Invitrogen) for 15 min at 37 °C and 95% O$_2$, 5% CO$_2$ humid atmosphere. The cells were then washed twice and then resuspended in 2% FBS/PBS and 2 mM EDTA. Viability dye 7-AAD (1:250; BioLegend,

420403) was added for 10 min at RT. Cells were analyzed with BD FACS Aria Fusion. Mean fluorescence intensity of all viable cells was analyzed using BD FACS DIVA software.

## Bioenergetics

Oxygen consumption rates (OCR) and extracellular acidification rates (ECAR) were measured using Seahorse XFe96 Analyzer (Agilent), and a mitochondrial stress test was performed as previously described (Ludikhuize et al, 2021). Organoids were imaged before the assay, and a total area was determined using the OrgaQuant tool (Kassis et al, 2019) and then used for normalization.

## RNA sequencing

For bulk RNA sequencing, total RNA was isolated from stimulated organoids using RNeasy Micro Kit (Qiagen) and quantified using Qubit High Sensitivity Assay (Invitrogen). For scRNA-seq, stimulated organoids from eight wells were pooled and dissociated into single cells, bigger fragments removed by a 30-µm strainer, and 6000 cells were processed using the ChromiumNext GEMSingle Cell 3′Reagent Kits Dual Index Kit (10x Genomics). The concentration of the prepared library was analyzed, and fragment size confirmed using High Sensitivity DNA Kit (Agilent).

The process of bulk RNA sequencing was carried out at Novogene using an Illumina Novaseq HiSeq Pair-Ended 150 bp, with a sequencing depth of 9 G, equivalent to 30 million reads for each sample. Similarly, scRNA-seq was executed, achieving a sequencing depth of 90 G for every sample.

## Bulk RNA-seq data analysis

Bulk RNA-seq data was preprocessed using the nf-core/rnaseq pipeline (version 3.8.1) (Patel et al, 2022). In brief, reads were aligned using STAR (Dobin et al, 2013) to reference genome version GRCh38 with genome annotation version GRCh38 (release 106, Ensembl), and gene expression was quantified using Salmon (Patro et al, 2017).

Differentially expressed genes were identified using DESeq2 (Love et al, 2014). Genes were included in differential expression analysis if they were detected with more than 1 count-per million in at least $n$ samples, where n is the number of samples in the smallest group of samples in the comparison. An adjusted $P$ value (FDR) < 0.1 was applied as threshold to consider genes significantly differentially expressed.

Differential expression of organoids treated with SLE or control serum: If samples from multiple organoid donors were available, organoid donor was included in the generalized linear model in addition to serum treatment as explanatory variable, thereby accounting for variation from different organoid donors. Contrasts were extracted for serum treatment as the main effect of interest (SLE vs. control).

Differential expression of human descending colon biopsies of SLE or control subjects: Gender was included in the generalized linear model in addition to disease status as explanatory variable, thereby accounting for variation from gender. Contrasts were extracted for disease status as the main effect of interest (SLE vs. control).

**The paper explained**

**Problem**

There is growing evidence that many systemic pathologies present with increased gut leakiness and dysbiosis. This is also the case for systemic lupus erythematosus, an autoimmune disease that affects the hematologic system. However, little is known about the cause and the impact on the intestinal epithelial barrier. We sought to develop an in vitro model that mimics the influence of disease-specific cytokine profile on the gut by stimulating organoids with patient serum.

**Results**

We could show that our novel near-physiologic model contains all major cell types of the intestine making it a valuable tool to study effects on cell-type composition and barrier function. Our results show that SLE serum stimulation resulted in a reduction of secretory cells that manifested in a change in mucus composition. The barrier function was further impaired by an increased permeability and altered mitochondrial function. The observed changes were type I interferon-dependent and SLE-specific.

**Impact**

This study provides novel insights into the impact of SLE on the intestinal epithelial barrier. The effect on the mucus layer has the potential to affect the gut microbiome, which highlights the complex interconnectivity within the intestine. Our in vitro model opens many scientific avenues to unravel underlying mechanisms and to develop novel therapeutic approaches. Furthermore, it can be utilized for other diseases to understand the effect of systemic immune responses on the intestinal epithelial barrier.

Gene set enrichment: Enrichment of terms was calculated using a hypergeometric test (hypeR package (Federico and Monti, 2020)), using gene sets available from the Molecular Signatures Database (MSigDB (Liberzon et al, 2011; Dolgalev, 2022)). Terms were considered significantly enriched with an adjusted *P* value (FDR) < 0.05 threshold. To visualize the similarity of terms based on shared genes, a distance matrix was calculated containing the pairwise distances between terms. As distance metric, binary distance was used, i.e., the proportion of non-shared genes amongst the union of genes from a given pair of terms. The resulting distance matrix was used as input for multidimensional scaling to visualize the similarity of terms in two dimensions.

## IFN score calculation

IFN score was calculated as z-score-based standardized score (Kim et al, 2018), using the transformed count data obtained through variance stabilizing transformation (DESeq2) as input and the following genes as interferon signature genes (IFNSG): ADAR, BST2, HLA-A, HLA-B, HLA-C, IFI27, IFI35, IFI6, IFIT1, IFIT3, IFIT5, IFITM1, IFITM3, IRF7, IRF9, ISG15, MX1, MX2, OAS1, OAS2, OAS3, OASL, PSMB8, SAMHD1, STAT1, USP18, XAF1.

## scRNA-seq data analysis

scRNA-seq data was preprocessed using the nf-core/scrnaseq pipeline (version 2.2.0) (Peltzer et al, 2023). In brief, read alignment, filtering and counting was performed using cellranger using a reference package created from reference genome version

GRCh38 with genome annotation version GRCh38 (release 106, Ensembl). Further analysis steps were performed using Seurat (Hao et al, 2021). Genes that were detected in fewer than four cells (from the whole dataset consisting of 34,255 cells) were excluded. Cells were included in further analysis if the number of detectable genes was more than 200, the percentage of mitochondrial reads was less than 25%, and the percentage of ribosomal reads was more than 5%. Doublets were excluded using DoubletFinder (McGinnis et al, 2019).

For further downstream analyses using the Principal Components (e.g., UMAP, nearest-neighbor graph construction for clustering), the first 60 Principal Components were considered. For further downstream analyses with a parameter for the number of nearest neighbors (e.g., UMAP, nearest-neighbor graph construction for clustering), a value of 300 and 50 was used for the analysis on all cells and Goblet cells, respectively. For clustering, a cluster resolution of 1 and 0.5 was used for the analysis on all cells and Goblet cells, respectively. For differential expression between two groups of cells, Wilcoxon rank-sum test was used and genes with an adjusted *P* value < 0.1 (Bonferroni correction) were called significant. Trajectory analysis was performed using the Slingshot package (Street et al, 2018) with SC1 set to be the cluster of origin (parameter "start.clus").

## Statistical analyses

Data are represented as mean with standard deviation except for organoid size analysis where data is shown as median with quartiles and cell cycle distribution, where data are shown as mean with standard error. If not stated otherwise organoid line II was used for analysis. Analyses were performed with unpaired *t* test or ANOVA with Holm–Šidák's multiple comparisons test. In the case of organoid size an average of 100 organoids was calculated, and the resulting values were compared using chi-square test. For the LDH and permeability assay as well as for IFN-α/β reporter cell line, FCGBP and EdU analysis a Kolmogorov–Smirnov test was used as the data was not following a Gaussian distribution. To calculate statistical significance for the cytokine measurements undetected cytokines were considered as half of the detection limit. If not stated with exact *P* values, *P* values below 0.05 were considered significant. Detailed information on the statistical test used can be found in the respective figure legend. All data points shown as technical replicates refer to individual wells of organoids stimulated with the same serum. All statistical analysis not connected to bulk or scRNA-seq were performed in GraphPad Prism 9 Software (GraphPad Software, Inc.). Datasets were analyzed for normal distribution using D'Agostino and Pearson test, and Anderson–Darling test. Outlier were identified using the ROUT outlier test (*Q* = 1%). All experiments were randomized and blinded whenever possible. No statistical methods were used to predetermine sample size.

## Data availability

RNA sequencing data that support the findings of this study are in the process of uploading at the European Genome-Phenome Archive (EGA). Datasets: EGAD50000000018 (https://ega-archive.org/datasets/EGAD50000000018). Study: EGAS50000000012.

## Peer review information

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

## Acknowledgements

We deeply appreciate the substantial contributions to this study from individuals and institutions alike. First, our sincere thanks to Hugo de Jonge for his crucial donation of the R-Spondin 1 cell line. We also express our gratitude to Alina Winkelotte and Almut Schulze for their invaluable help with the Seahorse analysis. Stephan Schöler's expertly crafted data analysis scripts significantly expedited our research and clarified our findings. The quality of our work was greatly enhanced by the critical input and guidance from Christoph Becker and Daigen Xu. Our findings were deeply enriched by the insightful contributions from Alexandros Drainas and Mojca Frank-Bertoncelj. Rafael Sênos Demarco provided invaluable advice on mitochondria assays. Our thanks also go to Novogene, with their assistance with RNA sequencing, the Heidelberg Nikon Imaging Center at the University of Heidelberg, part of BioQuant, for providing necessary resources and facilities. Special recognition to the patients, Novabiosis, DTI foundation, and the I3PT Biobank for their collaboration. We particularly thank Joaquim Albiol, Fernando Mosteiro, and his dedicated team for their indispensable help. All contributions, regardless of size, profoundly impacted the success of this study, and we apologize for any unintentional omissions. The research was financially supported by Merck KGaA, Darmstadt, Germany, during the employment of MRD, IVH, SE, MS, SB and BS at the BioMed X Institute.

## Author contributions

**Inga Viktoria Hensel**: Conceptualization; Data curation; Formal analysis; Validation; Investigation; Visualization; Methodology; Writing—original draft; Writing—review and editing. **Szabolcs Éliás**: Formal analysis; Visualization; Writing—review and editing. **Michelle Steinhauer**: Investigation. **Bilgenaz Stoll**: Investigation. **Salvatore Benfatto**: Investigation. **Wolfgang Merkt**: Resources. **Stefan Krienke**: Resources. **Hanns-Martin Lorenz**: Resources. **Jürgen Haas**: Resources. **Brigitte Wildemann**: Resources. **Martin Resnik-Docampo**: Conceptualization; Formal analysis; Supervision; Funding acquisition; Investigation; Visualization; Writing—original draft; Project administration; Writing—review and editing.

## Disclosure and competing interests statement

The authors declare no competing interests.

# Expanded View Figures

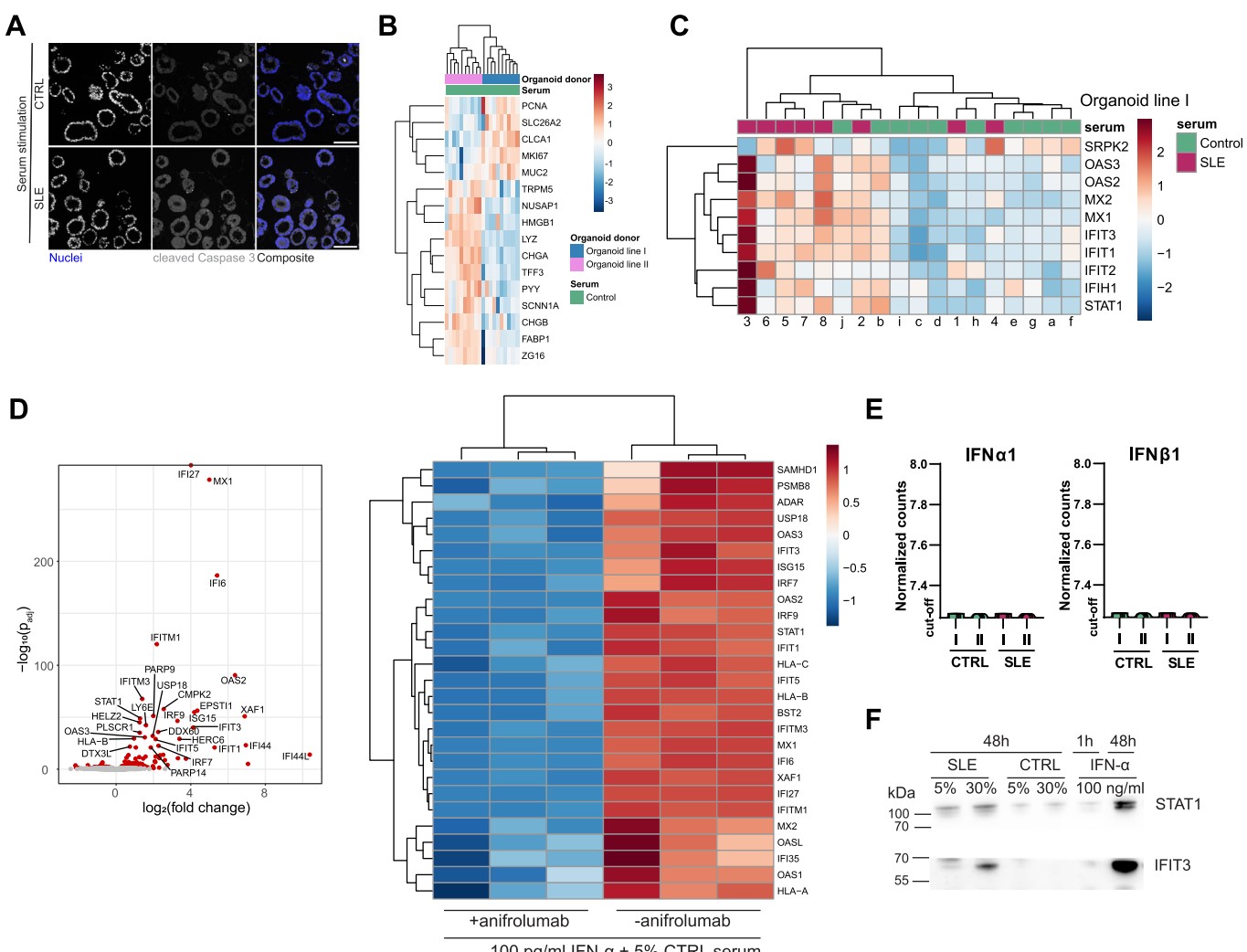

**Figure EV1. Expression profile of organoids stimulated with serum or with IFN-α in combination with anifrolumab.**

(A) Immunofluorescence of cleaved caspase 3 in either control (top) or SLE (bottom) serum-stimulated organoids. Nuclei marked by Hoechst33342 in blue and cleaved caspase 3 in gray, scale bars = 100 μm. (B) Heatmap with hierarchical clustering of various cell-type markers showing organoid line specific proliferation and differentiation dynamics, $n = 10$ control sera per organoid line I and II, respectively. (C) Heatmap with hierarchical clustering of selected IFN-inducible genes comparing SLE ($n = 8$) to control ($n = 10$) serum-stimulated organoids from organoid line I. (D) Volcano Plot showing DEGs (red) comparing organoids stimulated with IFN-α+control serum to additional anifrolumab treatment in organoid line II ($n = 3$ for each condition). Type I IFN-inducible genes are highlighted by the label. Corresponding heatmap with hierarchical clustering of 27 specific genes relevant in IFN-α/β signaling. (E) Analysis of normalized counts for IFN-α1 and IFN-α2 expression in SLE ($n = 8$) and control ($n = 10$) serum-stimulated organoids of line I and II. (F) Western blot analysis of STAT1 and IFIT3 expression in IFN-α, SLE and control serum-stimulated organoids after 48 h. Each sample represents two pooled wells of organoids stimulated with serum from one donor.

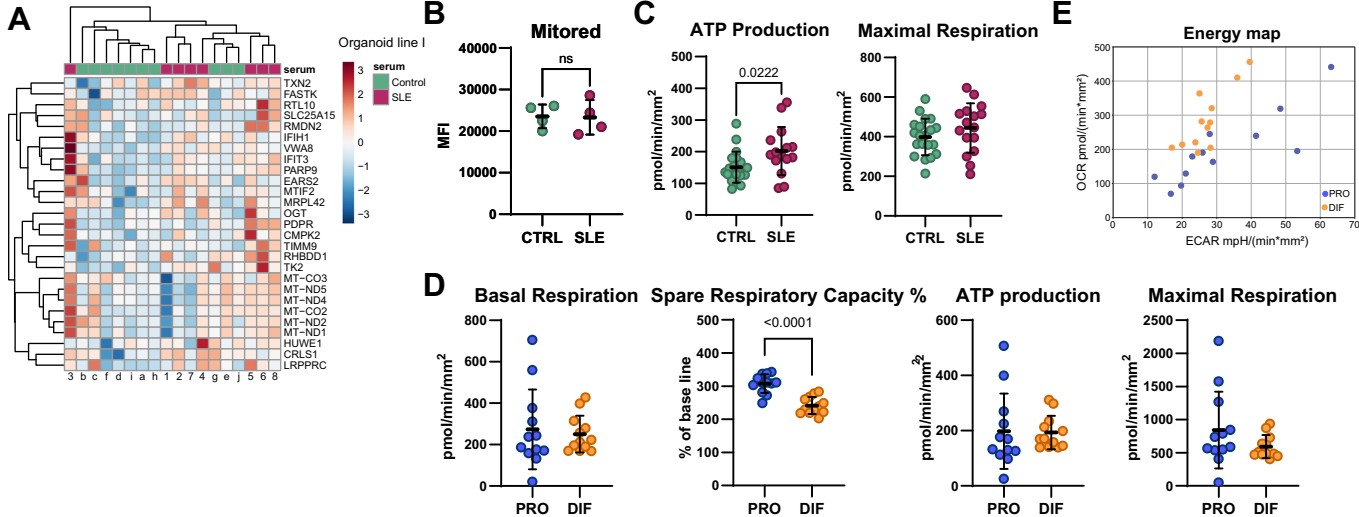

**Figure EV2. Alterations of mitochondrial function are SLE serum specific.**

(A) Heatmap with hierarchical clustering of mitochondrial genes comparing SLE to control serum-stimulated organoids from organoid line I. (B) Flow cytometry analysis of control and SLE serum-stimulated organoids stained with mitored. Each dot represents one well of organoids dissociated into single cells and analyzed as mean fluorescence intensity (MFI); $n = 4$ serum for each condition. Data are represented as mean ± SD, and statistical significance determined by unpaired $t$ test. (C) Seahorse assay showing ATP production (left) and maximal respiration (right) comparing control (green) and SLE (pink) serum-stimulated organoids. Data are represented as mean ± SD, and statistical significance determined by unpaired $t$ test. Each dot corresponds to one well of organoids stimulated with $n = 10$ control serum samples or $n = 8$ SLE serum samples. Each serum stimulation was analyzed with $n = 2$ technical replicates. (D) Seahorse assay showing basal respiration, relative spare respiratory capacity, ATP production and maximal respiration comparing organoids with a proliferative cell-type composition (PRO, blue, $n = 12$) to organoids containing additionally differentiated cells (DIF, orange, $n = 12$). Data are represented as mean ± SD, and statistical significance determined by unpaired $t$ test. (E) Energy map illustrating the metabolic profile of organoids in a mainly proliferative state (blue) or additionally containing differentiated cells (orange).

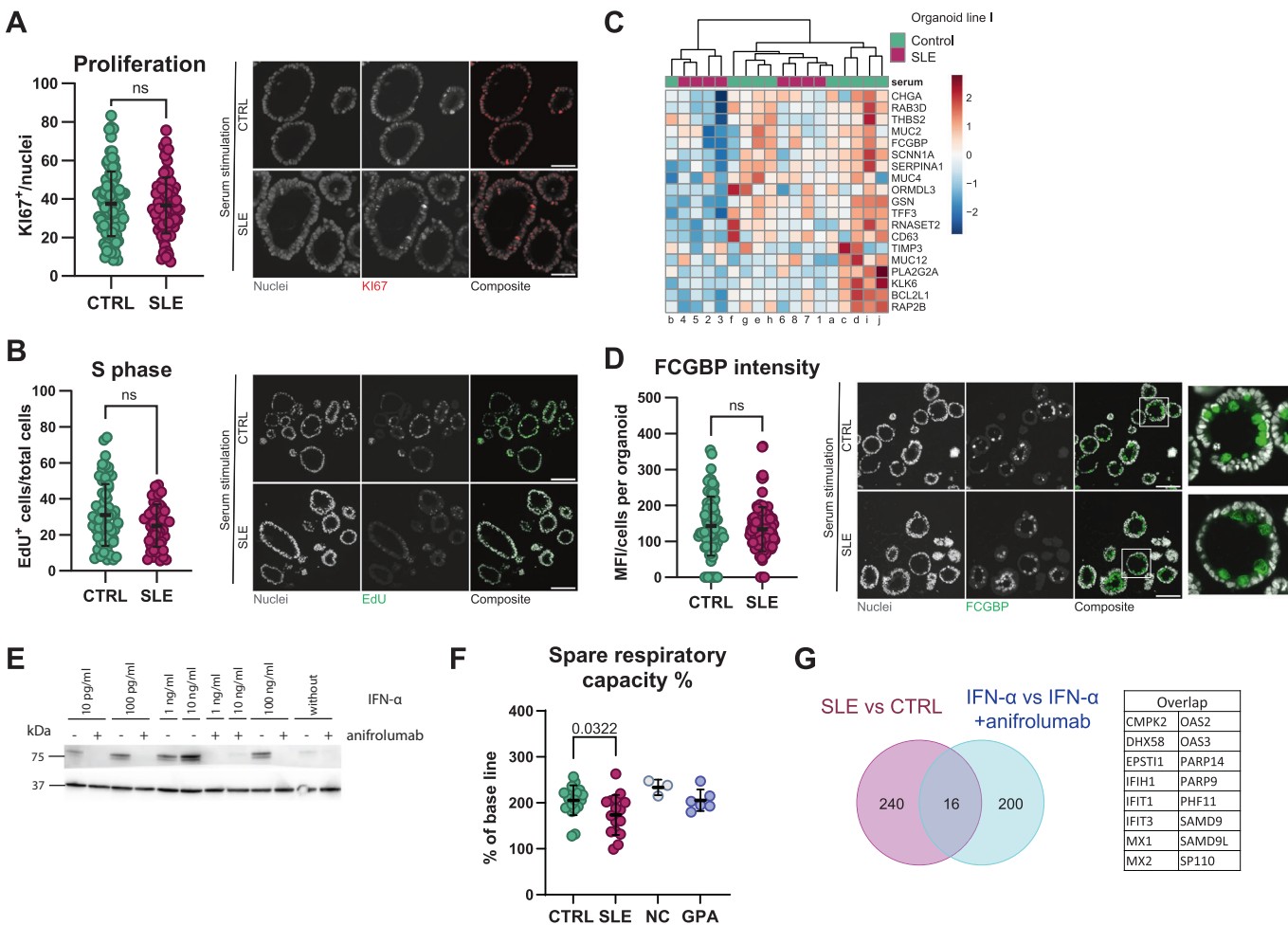

**Figure EV3. Proliferation is unaltered while mitochondrial changes are distinctive to SLE serum stimulation.**

(A) Quantification of KI67 positive cells per organoid normalized to the number of total nuclei per organoid. Each dot corresponds to a single organoid (organoid line II) stimulated with control ($n = 98$) or SLE serum ($n = 78$). Data are represented as mean ± SD, and statistical significance determined by unpaired $t$ test. Immunofluorescence of KI67 in either control (top) or SLE (bottom) serum-stimulated organoids. Nuclei marked by Hoechst33342 in gray and KI67 in red. Scale bar = 50 μm. (B) Quantification of EdU (5-Ethynyl-2′-deoxyuridine) positive cells per organoid normalized to the number of nuclei per organoid. Each dot corresponds to a single organoid (organoid line II) stimulated with control ($n = 64$) or SLE serum ($n = 48$). Data are represented as mean ± SD, and statistical significance determined by Kolmogorov–Smirnov test. Immunofluorescence of EdU in either control (top) or SLE (bottom) serum-stimulated organoids. Nuclei marked by Hoechst33342 in gray and EdU in green. Scale bar = 100 μm. (C) Heatmap with hierarchical clustering of downregulated genes associated with the gene set "Secretory Vesicle" and "Mucus" comparing SLE to control serum-stimulated organoids from organoid line I. (D) Quantification of FCGBP mean intensity per organoid normalized to the number of total nuclei per organoid. Each dot corresponds to a single organoid (organoid line II) stimulated with control ($n = 79$) or SLE serum ($n = 126$). Data are represented as mean ± SD, and statistical significance determined by Kolmogorov–Smirnov test. Insets showcase enlarged areas, scale bar = 100 μm. (E) Western blot analysis of STAT1 phosphorylation in organoids stimulated for 48 h with increasing concentrations of IFN-α2 with or without anifrolumab treatment to validate its inhibitory effect. (F) Seahorse assay showing relative spare respiratory capacity comparing control (green) and SLE (pink) serum-stimulated organoids to unstimulated (NC, gray) and GPA (granulomatosis with polyangiitis) serum (violet) stimulated organoids. Data are represented as mean ± SD, and statistical significance determined by one-way ANOVA with Holm–Šidák's multiple comparisons test. Each dot corresponds to one well of organoids stimulated with control or SLE serum ($n = 10$) or GPA serum ($n = 3$). Each serum stimulation was analyzed with $n = 2$ technical replicates except NC with $n = 3$ technical replicates. (G) Venn diagram showing the overlap of DEGs resulting from stimulation with SLE compared to control serum and IFN-α+control serum compared to control serum. The overlapping genes are represented in the table.

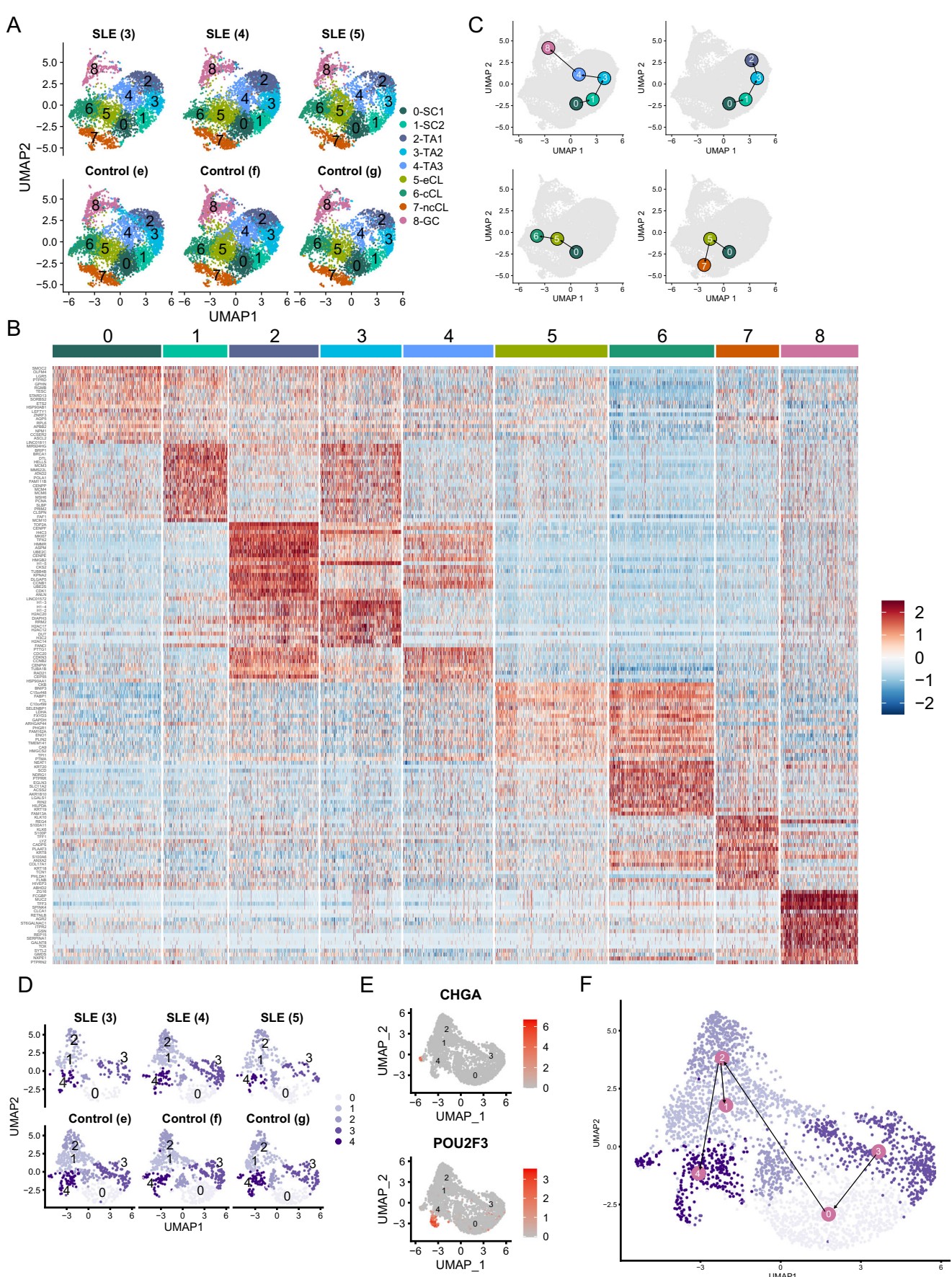

◄ **Figure EV4.  scRNA-seq data showing sample specific clustering, heatmap and developmental trajectories.**

(A) UMAP analysis showing the cluster distribution for each analyzed sample. (B) Heatmap representing the top 20 DEG per identified cluster. (C) UMAP plot illustrating the different predicted cellular developmental trajectory using Slingshot. (D) UMAP analysis showing subclustering of goblet cells for each analyzed sample. (E) UMAP plot illustrating the expression of CHGA and POU2F3 in goblet cell subclusters. (F) UMAP plot illustrating the different predicted cellular developmental trajectory within the goblet cell subcluster using Slingshot. All plots except for (A and D) show the data from all analyzed cells independent on the serum source.

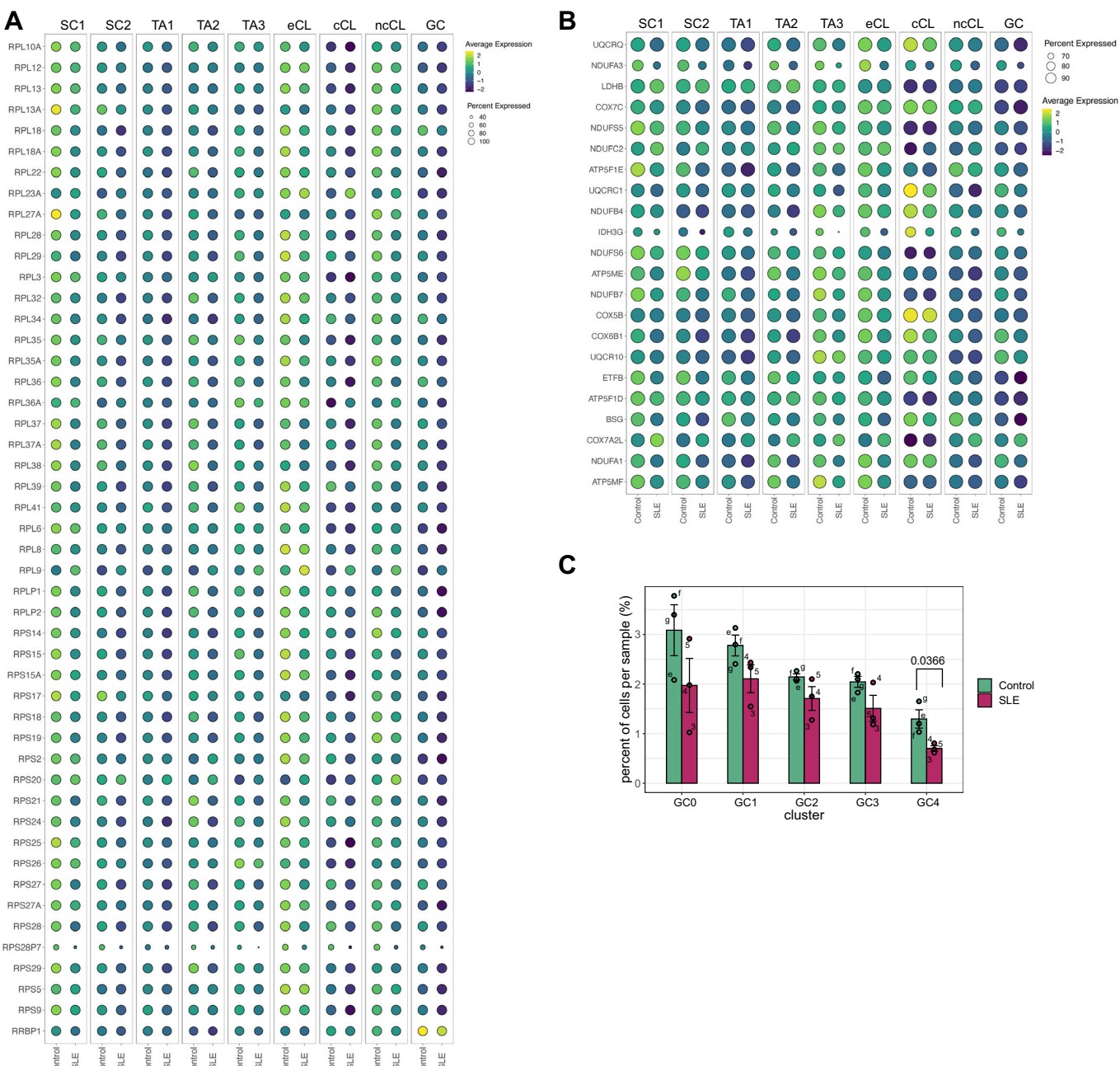

**Figure EV5. Effect of SLE serum stimulation on ribosomal, mitochondrial gene expression and goblet cell subtypes.**

(A) Dot plot representing the average expression of ribosomal genes across all identified cell types. (B) Dot plot representing the average expression of genes connected to oxidative phosphorylation across all identified cell types. (C) Analysis of the distributions of goblet cell subclusters within the total analyzed cells of SLE (pink, $n = 3$) and control (green, $n = 3$) serum-stimulated organoids. Data are presented as mean with ±SD. Unpaired $t$ test was performed. (A, B) The size and color of each dot represent the cell percentage and average gene expression, respectively.

