## [Peer Review File · EMBO Molecular Medicine]

SLE Serum Induces Altered Goblet Cell Differentiation and Leakiness in Human Intestinal Organoids

Inga Hensel, Szabolcs Éliás, Michelle Steinhauer, Bilgenaz Stoll, Salvatore Benfatto, Wolfgang Merkt, Stefan Krienke, Hanns-Martin Lorenz, Juergen Haas, Brigitte Wildemann, and Martin Resnik-Docampo

DOI: [10.15252/emmm.202318796](https://doi.org/10.15252/emmm.202318796)

Corresponding author(s): *Martin Resnik-Docampo (resnik@bio.mx)* , *Inga Hensel (hensel@bio.mx)*

Review Timeline:	Transfer from Review Commons:	4th Oct 23
	Editorial Decision:	31st Oct 23
	Revision Received:	16th Nov 23
	Editorial Decision:	4th Dec 23
	Revision Received:	21st Dec 23
	Accepted:	5th Jan 24

Editor: *Lise Roth*

Transaction Report: This manuscript was transferred to EMBO reports following peer review at Review Commons.

**Review
COMMONS**

Review #1

1. Evidence, reproducibility and clarity:

Evidence, reproducibility and clarity (Required)

Authors used organoid technology to study the effects of the serum from lupus patients on intestinal epithelium. By culturing organoids derived from human colon crypts, they specifically determined the response of epithelial cells to inflammatory mediators present in lupus serum. Using bulk and scRNA-seq, authors found that secretory cells function and differentiation were impaired as well as the mitochondrial metabolism. These effects were shown to be mediated by type 1 interferon in combination with other pro-inflammatory cytokines present in lupus serum.

The reduction of mucus secretion after SLE-serum treatment and the downregulation of tight junctions' genes seem to indicate an increased permeability of the epithelial barrier, thus it would be interesting to determine the expression and distribution of tight junction proteins and to test in the organoids whether the paracellular permeability is increased upon SLE-serum treatment. These analyses will give a functional result of this in vitro model.

If the organoids take a few days to culture and the material is available, the measurement of paracellular permeability may take no more than 2 weeks. It is true that they will need a microneedle to inject the FITC-Dextran 4K into the organoids and record the images for 24h.

I would like to know which of the donor's cells were used from figure 2 on and why.

The bioinformatics analyses using gene expression data and scRNASeq were well done. No comments.

2. Significance:

Significance (Required)

For the field of autoimmunity, to study the crosstalk between the systemic response and the gut epithelium response results quite important as the increased permeability of the gut epithelial barrier has been suggested to fuel the systemic inflammation in lupus. However, as the author mention, there is not enough information about the interaction of epithelial cells and the systemic inflammatory mediators in lupus. This

system can be useful to determine a personalized treatment for patients by testing the effect of individual serum on organoids. Moreover, the use of organoids can be extended to study the gut epithelium response in other autoimmune diseases mediated by type 1 interferon.

Increased permeability of the gut epithelial barrier has been related with lupus development. In humans, it is not known whether it is a cause or consequence, but in lupus mouse models it has been demonstrated that there is a reduction of the systemic autoimmune response concomitant with a reduction of gut permeability. The authors have validated an in vitro model that can be used to study how gut epithelium is affected by systemic inflammatory mediators and that will help to develop novel therapeutic approaches or personalize treatments.

Interest stakeholders: Clinical and basic researchers in autoimmunity, gastroenterology, and rheumatology.

My field of expertise is systemic and organ-specific autoimmunity at cellular and molecular level. My work covers autoimmunity and gut microbiota. I study how B cells regulate the microbiota composition and how that microbiota impacts gut permeability and inflammation in mouse lupus models. On the other hand, the bioinformatics analyses are well-done for both bulk RNASeq and scRNASeq.

3. How much time do you estimate the authors will need to complete the suggested revisions:

Estimated time to Complete Revisions (Required)

(Decision Recommendation)

Less than 1 month

Yes

Review #2

1. Evidence, reproducibility and clarity:

Evidence, reproducibility and clarity (Required)

The group of Dr. Resnik-Docampo provides a very elegant study on two patient lines for SLE. The study is definitely very interesting and opens many scientific avenues that are worthy of being explored further.

Major comments:

- Barrier integrity or its alterations can be tested in organoids with specific dyes, I feel this would give definitive proof of concept.
- In supplementary figure 1 a caspase 3 staining is presented, please show a positive control for caspase 3 staining on organoids or alternatively use a different method to prove no differential cell death.
- Serum from SLE patients reduces drastically Edu positivity, it would be interesting to see a clonogenicity assay to see whether this reflects on reduced stem cell clonogenic potential
- Goblet cells in the colon are very heterogeneous, which subpopulations of goblet cell are reduced? how does this affect mucus composition?

Minor comments:

- Please provide an hypothesis on how mitochondrial alterations are linked to altered lineage progeny of stem cells. This should be discussed more in depth.
- Many antimicrobial peptides are changed, this could reflect on microbiome composition as well as mucus composition and properties, which I am sure will be the topic of future studies. This should be discussed more in depth.

2. Significance:

Significance (Required)

The study is useful for both broad and specialised audiences. The findings are interesting and of relevance to the field of SLE, gut epithelial biology. The strength of the manuscript is that it opens many scientific avenues, its weakness is that they are not mechanistically dissected to the fullest rendering the study a bit descriptive. Nonetheless, I consider positively the manuscript after a minor revision given the major message of the paper can be proven.

3. How much time do you estimate the authors will need to complete the suggested revisions:

Estimated time to Complete Revisions (Required)

(Decision Recommendation)

Between 1 and 3 months

Yes

Review #3

1. Evidence, reproducibility and clarity:

Evidence, reproducibility and clarity (Required)

Inga Viktoria Hensel et al. used colon organoid to study the impact of lupus patients' serum on gut epithelial barrier. The exposure of SLE serum on colon organoids increased gene expression related to cell cycle, chromosome organization,

mitochondrial function as well as interferon signaling, but downregulated that related to secretion, cytoskeleton, and anchoring junctions of the cells. Higher type I IFN in the SLE serum and unregulated interferon signature genes post stimulation suggest a potential role of type I interferon in this process. The addition of a type 1 interferon receptor (IFNAR1) antagonist, Anifrolumab, blocked the stimulation function of SLE serum but the combination of IFN-2 α and control serum failed to recapitulate the results from SLE serum, suggesting that more than one cytokine was involved. SLE serum exposure altered metabolic profiles of organoids with a significant increase of basal respiration and ATP production. Stimulating organoid with SLE serum confirmed an alteration in cell differentiation with a loss of secretory lineage. scRNA-seq analysis revealed that colon organoid had all major cell types from colon in vivo. SLE serum stimulation shifted cell differentiation with decreased number of goblet cells and downregulated mucin, AMP and other components that were required for gut barrier integrity.

Finally, the authors performed a gene expression analysis of colon biopsies derived from SLE patients and healthy controls. While the authors should be commended to attempt a validation of the results obtained with organoids, the small sample size and patient heterogeneity prevented a statistical analysis. Some genes involved in absorption and ion transport as well as secretory lineage showed a similar trend with organoid assay, suggesting that colon organoids may be a good tool for future studies. However, it is noticeable that the biopsies from SLE patients did not show the IFN signature and the decreased in Muc2 expression, which dominated the gene signature of organoids exposed to SLE serum. There is no information about the disease activity of the SLE patients, as well as their IFN activity, which makes difficult to interpret these results.

****Specific concerns:****

1. Line 107, Why did The authors use 72 hours post treatment. Are other timepoints available and have similar results?
2. Figure 1D, how do the authors explain the heterogeneity among SLE samples (2, 3, 4, 5) on organoid line II? These samples do not seem to correlate with cytokine levels shown in Fig. 2. This issue may be worth exploring further, such as correlation between cytokine levels and gene expression.
3. Line 144, the 2 outlier SLE serum samples are not same between organoid lines with NO. 1&5 in Organoid line II and with NO. 1&4 in Organoid line I. The statement is misleading.
4. Line 169, IFN-a2 and IL-6 are not significantly different.
5. Line 179-180, Reduced fitness of organoids exposed to SLE serum is an

overstatement. It was not directly tested, and there is no difference in apoptosis.

6. Line 242-243: SLE serum stimulation induced MUC2 high expression in Organoid II but lower level in organoid I (Figure 4B & Figure S4C). This is a major discrepancy that needs to be addressed.

7. Line 251-252: How do authors make sure that "we were facing an effect on the differentiation process rather than cell type loss"?

8. Line 259, How about apoptosis gene levels here?

9. Line 290 : It has been shown that the response of colon explants to IFN- α was variable among donors

(<https://www.sciencedirect.com/science/article/pii/S2352345X16301084#undfig1>).

This study should be cited. Was the response to IFN α tested on both organoids?

10. Line 305-307: Is single cell sequencing from single organoid line or from combined? Do two organoid lines show different distributions?

2. Significance:

Significance (Required)

While there is mounting evidence of an altered intestinal barrier integrity in SLE patients, there is little insights in the mechanisms. Using colon organoids is a novel approach with great potentials to investigate this issue. The strongest signature found by the authors was type IFN, which is indicates that the colon epithelial cells respond in a similar manner to other cell types. The secretory and absorption genes are of potential greater interest to unraveling mechanisms to gut alteration in SLE.

This study is of interest for audiences interested in lupus basic and clinic research, as well as investigators working of gut barrier integrity.

3. How much time do you estimate the authors will need to complete the suggested revisions:

Estimated time to Complete Revisions (Required)

(Decision Recommendation)

Between 1 and 3 months

4. *Review Commons* values the work of reviewers and encourages them to get credit for their work. Select 'Yes' below to register your reviewing activity at Web of Science Reviewer Recognition Service (formerly Publons); note that

the content of your review will not be visible on Web of Science.

No

Full Revision

Manuscript number: RC- 2023-02065

Corresponding author(s): Martin, Resnik Docampo

1. General Statements [optional]

This section is optional. Insert here any general statements you wish to make about the goal of the study or about the reviews.

In response to the feedback received from reviewers, we have taken several steps to further refine and enhance our manuscript. In this revised version we addressed every single point that the reviewers raised. Minor edits were made regarding more concise wording and/or phrasing of our statements and corrections of misspelling. As major changes, we added the data from two more experiments which were suggested by the reviewers:

1. Permeability assay which is shown in Fig. 1H.
2. Further analysis of mucus composition which is shown in F Fig. 4C and Suppl. Fig. 4D.

The new experiments gave us valuable new insights which made it necessary to adjust the title of our manuscript.

In the revised version, we've integrated suggested experiments, enhancing our manuscript's depth and quality. The functional validation of intestinal permeability in our model provides robust evidence of the link between SLE and intestinal barrier integrity. This addition emphasizes the translational significance of our study, highlighting its relevance for researchers in both fundamental and clinical lupus research, as well as for specialists in intestinal biology.

For clarity:

In this document, reviewers' comments are highlighted in blue, our responses are in black, and revisions to the manuscript, including corrections and experiments suggested by the reviewers, are marked in red.

Reviewer #1 (Evidence, reproducibility and clarity (Required)):

Authors used organoid technology to study the effects of the serum from lupus patients on intestinal epithelium. By culturing organoids derived from human colon crypts, they specifically determined the response of epithelial cells to inflammatory mediators present in lupus serum. Using bulk and scRNA-seq, authors found that secretory cells function and differentiation were impaired as well as the mitochondrial metabolism. These effects were shown to be mediated by type 1 interferon in combination with other pro-inflammatory cytokines present in lupus serum.

The reduction of mucus secretion after SLE-serum treatment and the downregulation of tight junctions' genes seem to indicate an increased permeability of the epithelial barrier, thus it would be interesting to determine the expression and distribution of tight junction proteins and to

Full Revision

test in the organoids whether the paracellular permeability is increased upon SLE-serum treatment. These analyses will give a functional result of this in vitro model.

If the organoids take a few days to culture and the material is available, the measurement of paracellular permeability may take no more than 2 weeks. It is true that they will need a microneedle to inject the FITC-Dextran 4K into the organoids and record the images for 24h.

- We thank the reviewer for suggesting this experiment. Reviewer #2 had the same suggestion. The results obtained elevate our overall results and the quality of our research.
- Tight junction protein expression is cell-type dependent as shown in literature^{1,2} and in our scRNA-seq analysis (Suppl. Table 7). Additionally, we observed that the expression changes of ZO-1(TJP1; Fig. 7A) are accumulated in colonocytes. That might be the reason we could not detect significant changes in organoid sections with staining for tight junction proteins ZO-1 and Occludin and analyzing all cell types. The data of the permeability assay however was able to show that the function of the tight junctions is altered. If this change is caused by changes in protein abundance as suggested by reported transcriptomic changes remains to be elucidated in future research using single-cell proteomic approaches.
- For the permeability assay we stimulated organoid monolayers for 72h with SLE or control serum. The measured translocation of FITC showed an increased barrier leakiness in SLE compared to control condition. These functional results not only support our findings presented in our study that the epithelial barrier is altered upon SLE serum stimulation, but they also provide the definitive proof of concept highlighting the crucial connection between SLE and intestinal barrier integrity. Furthermore, it shows that changes on transcriptional level and alterations in cell type composition translate into a barrier dysfunction which could have potentially detrimental effects *in vivo*. The data has been added to Figure 1H, line 722-731; 746-751.

I would like to know which of the donor's cells were used from figure 2 on and why.

- Organoid line I was used for the 24h stimulation and for the 72h stimulation whereas organoid line II was used for the 72h stimulation only. Due to limitations in serum availability we had to limit the experiments that exceeded the initial transcriptomic analysis to one organoid line. After confirming that the serum was affecting both organoid lines, we continued using organoid line II.

The bioinformatics analyses using gene expression data and scRNASeq were well done. No comments.

Reviewer #1 (Significance (Required)):

For the field of autoimmunity, to study the crosstalk between the systemic response and the gut

Full Revision

epithelium response results quite important as the increased permeability of the gut epithelial barrier has been suggested to fuel the systemic inflammation in lupus. However, as the author mention, there is not enough information about the interaction of epithelial cells and the systemic inflammatory mediators in lupus. This system can be useful to determine a personalized treatment for patients by testing the effect of individual serum on organoids. Moreover, the use of organoids can be extended to study the gut epithelium response in other autoimmune diseases mediated by type 1 interferon.

Increased permeability of the gut epithelial barrier has been related with lupus development. In humans, it is not known whether it is a cause or consequence, but in lupus mouse models it has been demonstrated that there is a reduction of the systemic autoimmune response concomitant with a reduction of gut permeability. The authors have validated an *in vitro* model that can be used to study how gut epithelium is affected by systemic inflammatory mediators and that will help to develop novel therapeutic approaches or personalize treatments.

Thank you for your insightful comments. We are encouraged to see that the reviewer has accurately grasped the core purpose and implications of our research. The intricate relationship between gut epithelial barrier permeability and lupus development is indeed vital for expanding our scientific knowledge and for future therapeutic breakthroughs. We are happy that the overall message we aimed to convey with our research was well captured and appreciated by the reviewer. We believe our *in vitro* model will serve as a foundation for further detailed studies and for the development of therapeutic strategies in this field. Your acknowledgment of our work inspires us to persist in our research efforts.

Interest stakeholders: Clinical and basic researchers in autoimmunity, gastroenterology, and rheumatology.

My field of expertise is systemic and organ-specific autoimmunity at cellular and molecular level. My work covers autoimmunity and gut microbiota. I study how B cells regulate the microbiota composition and how that microbiota impacts gut permeability and inflammation in mouse lupus models.

On the other hand, the bioinformatics analyses are well-done for both bulk RNASeq and scRNASeq.

Reviewer #2 (Evidence, reproducibility and clarity (Required)):

The group of Dr. Resnik-Docampo provides a very elegant study on two patient lines for SLE. The study is definitely very interesting and opens many scientific avenues that are worthy of being explored further.

Major comments:

-Barrier integrity or its alterations can be tested in organoids with specific dyes, I feel this would give definitive proof of concept.

- We thank the reviewer for the suggestion. Reviewer #1 had the same suggestion. The results obtained elevate our overall results and the quality of our research.
- For the permeability assay we stimulated organoid monolayers for 72h with SLE or control serum. The measured translocation of FITC showed an increased barrier leakiness in SLE compared to control condition. These functional results not only support our findings presented in our study that the epithelial barrier is altered upon SLE serum stimulation, but they also provide the definitive proof of concept highlighting the crucial connection between SLE and intestinal barrier integrity. Furthermore, it shows that changes on transcriptional level and alterations in cell type composition translate into a barrier dysfunction which could have potentially detrimental effects *in vivo*. The data has been added to Figure 1H, line 722-731; 746-751.

-In supplementary figure 1 a caspase 3 staining is presented, please show a positive control for caspase 3 staining on organoids or alternatively use a different method to prove no differential cell death.

- Due to the processing of the organoids it is difficult to have a positive control. However, we can confirm that the used cleaved Caspase 3 antibody is able to detect cell death by the following staining. There are positive cells in the center of the organoid where dead cells accumulate. Additionally in Figure 1C we show with a cytotoxicity assay measuring LDH release that there is no significant difference between both groups. Furthermore, brightfield imaging showed no obvious differences and the DEGs show also no evidence of increased cell death.

-Serum from SLE patients reduces drastically Edu positivity, it would be interesting to see a clonogenicity assay to see whether this reflects on reduced stem cell clonogenic potential

- We agree. However, this analysis goes beyond the time and material limitations we have in our project.

-Goblet cells in the colon are very heterogeneous, which subpopulations of goblet cell are reduced? how does this affect mucus composition?

- We see that subpopulation GC4 is significantly reduced upon SLE serum stimulation (Fig. 7C and Suppl. Fig. 7C). Overall, we see a trend of a general reduction of all GC subpopulation and an indication that there might be also a shift in subtype abundance. We would need to increase the study population to be able to draw any conclusions on how exactly the GC subpopulations change. So far not much is known about how the different GC subpopulations affect mucus composition. This field is completely understudied especially in the human intestine. Future projects should focus on mucus composition and how it is changed with changes of the subpopulations (even under physiological conditions).
- We thank the reviewer for the question about the mucus composition. We included the analysis of FCGBP protein abundance (Fig. 4C and Suppl. Fig. 4D; line 245-247) in our study. The increased FCGBP protein abundance upon SLE serum stimulation which is in line with our transcriptomic changes complements our data and supports our hypothesis that the mucus composition is altered. This new data improves the quality of our research.

Minor comments:

- Please provide an hypothesis on how mitochondrial alterations are linked to altered lineage progeny of stem cells. This should be discussed more in depth.

- It is known that cells in the crypt compartment that undergo rapid division depend on glycolysis for ATP production once the cell differentiates it switches to oxidative phosphorylation.³ *In vitro* it has been shown that differentiation coincides with the switch to oxidative phosphorylation.⁴ There is a complex interplay between Notch signaling and FoxO transcription factors which is driving differentiation and cell fate decision.⁵ Especially the differentiation towards the secretory lineage highly depends on the metabolic switch from glycolysis to oxidative phosphorylation. Furthermore, even mucus secretion itself is dependent on oxidative phosphorylation.⁶ This is in line with our differentiation experiment where less oxidative phosphorylation coincided with the absence of goblet cells. We hypothesize that the changed cellular composition upon SLE serum stimulation is at least partly reflected in the altered mitochondrial function. If the decrease in the secretory lineage alone can explain the seen mitochondrial changes needs to be further elucidated. While a detailed examination at single-cell level analysis of mitochondrial function might be able to answer if they drive the altered cell differentiation, such an in-depth analysis is beyond the scope of this article and would be best addressed in a dedicated study on the topic. Within the scope of our analysis and considering the still limited knowledge of mitochondrial function in specific cell types of the human colon we included some more discussion (line 696-698 and 702-703).

-Many antimicrobial peptides are changed, this could reflect on microbiome composition as well as mucus composition and properties, which I am sure will be the topic of future studies. This should be discussed more in depth.

- We hypothesize that the alterations in antimicrobial peptide expression along with the seen changes in major mucus components would be translated into changes in mucus composition. Since the mucus serves as the niche for gut microbiota this could lead to changes in microbiome composition. It is of high interest to analyze the mucus composition of the stimulated organoids. Furthermore, assays analyzing the killing capacity of the potentially secreted antimicrobial peptides could help us to understand the relevance of the observed changes. We addressed this in line 714-715 and line 750-752.

Reviewer #2 (Significance (Required)):

The study is useful for both broad and specialized audiences. The findings are interesting and of relevance to the field of SLE, gut epithelial biology. The strength of the manuscript is that it opens many scientific avenues, its weakness is that they are not mechanistically dissected to the fullest rendering the study a bit descriptive. Nonetheless, I consider positively the manuscript after a minor revision given the major message of the paper can be proven.

Thank you for your constructive feedback. We are genuinely encouraged by your recognition of the utility and relevance of our study for both broad and specialized audiences in the fields of SLE and gut epithelial biology. Your acknowledgment resonates with our larger objectives: beyond merely exploring the specific connection between SLE and intestinal leakiness, our aim has been to create a methodological approach that illuminates novel avenues to study complex diseases. It is deeply heartening to realize that this overarching message and intent were clearly understood and agreed upon by the reviewer. We recognize and appreciate your insights into the strengths and areas for improvement of our manuscript, and we are committed to addressing the highlighted points to further enhance our contribution to the field.

Reviewer #3 (Evidence, reproducibility and clarity (Required)):

Inga Viktoria Hensel et al. used colon organoid to study the impact of lupus patients' serum on gut epithelial barrier. The exposure of SLE serum on colon organoids increased gene expression related to cell cycle, chromosome organization, mitochondrial function as well as interferon signaling, but downregulated that related to secretion, cytoskeleton, and anchoring junctions of the cells. Higher type I IFN in the SLE serum and unregulated interferon signature genes post stimulation suggest a potential role of type I interferon in this process. The addition of a type 1 interferon receptor (IFNAR1) antagonist, Anifrolumab, blocked the stimulation function of SLE serum but the combination of IFN-2 α and control serum failed to recapitulate the results from SLE serum, suggesting that more than one cytokine was involved. SLE serum exposure altered metabolic profiles of organoids with a significant increase of basal respiration and ATP production. Stimulating organoid with SLE serum confirmed an alteration in cell differentiation with a loss of secretory lineage. scRNA-seq analysis revealed that colon organoid had all major cell types from colon in vivo. SLE serum stimulation shifted cell differentiation with decreased number of goblet cells and downregulated mucin, AMP and other components that were required for gut barrier integrity.

Finally, the authors performed a gene expression analysis of colon biopsies derived from SLE patients and healthy controls. While the authors should be commended to attempt a validation of the results obtained with organoids, the small sample size and patient heterogeneity prevented a statistical analysis. Some genes involved in absorption and ion transport as well as secretory lineage showed a similar trend with organoid assay, suggesting that colon organoids may be a good tool for future studies. However, it is noticeable that the biopsies from SLE patients did not show the IFN signature and the decreased in Muc2 expression, which dominated the gene signature of organoids exposed to SLE serum. There is no information about the disease activity of the SLE patients, as well as their IFN activity, which makes difficult to interpret these results.

- We thank the reviewer for the questions and suggestions. They helped us to improve our manuscript and make it more concise for the reader.

Specific concerns:

1. Line 107, Why did The authors use 72 hours post treatment. Are other timepoints available and have similar results?

- We used two different time points, 24h and 72h. The nature of the organoid culture limits the total length of the experiment. Organoids can be cultured for a maximum of 5-7 days. Differentiation from the stem cell to the fully differentiated cell takes 5-7 days. Preliminary results showed that serum stimulation prior to day 2 led to a decrease in organoid survival, most likely because serum stimulation in general induces differentiation. We therefore chose 72h as the stimulation that mimics a chronic exposure as a differentiating cell would face *in vivo*. With this time span we were able to see manifestations in cell differentiation changes but avoided beginning cell death to prolonged culture. However, we also wanted to understand a more acute exposure to the serum. Therefore, we chose 24h as a second stimulation duration. With this time point we were able to detect initial changes in cell fate decision which was important in the interpretation of the accumulated effects seen after 72h.

2. Figure 1D, how do the authors explain the heterogeneity among SLE samples (2, 3, 4, 5) on organoid line II? These samples do not seem to correlate with cytokine levels shown in Fig. 2. This issue may be worth exploring further, such as correlation between cytokine levels and gene expression.

- The heterogeneity seen among the SLE samples most likely reflects the complex composition of the serum itself. A similar heterogeneity (PC1 axis) is also seen for the controls. The epithelial cells eventually will react to all contained factors showing an integrated response that makes it difficult to correlate to a single cytokine.

3. Line 144, the 2 outlier SLE serum samples are not same between organoid lines with NO. 1&5 in Organoid line II and with NO. 1&4 in Organoid line I. The statement is misleading.

- We corrected the statement according to the suggestions (Line 146-148).

4. Line 169, IFN- α 2 and IL-6 are not significantly different.

- The statement was rephrased (Line 166-174).

5. Line 179-180, Reduced fitness of organoids exposed to SLE serum is an overstatement. It was not directly tested, and there is no difference in apoptosis.

- The term 'reduced fitness' referred to the results seen in the mitochondrial stress test. We rephrased the parts to make the statement more concise (Line 182).

6. Line 242-243: SLE serum stimulation induced MUC2 high expression in Organoid II but lower level in organoid I (Figure 4B & Figure S4C). This is a major discrepancy that needs to be addressed.

- This discrepancy comes from the different differentiation dynamics observed in both organoid lines. Therefore, for the downstream analysis we considered the results from both lines to have a robust analysis. With the results from the 24h timepoint and the scRNA-seq which were performed with either of the lineages respectively, we can be certain that the overall seen effect on the secretory lineage is a valid finding (we addressed this in line 229-232).

7. Line 251-252: How do authors make sure that "we were facing an effect on the differentiation process rather than cell type loss"?

- We can exclude an increase of cell death since we did not see changes in LDH release (Fig. 1C) and cleaved Caspase 3 abundance (Suppl. Fig. 1A) when we compared both conditions. Additionally, the data from the 24h stimulation time point showed the reduction of transcription factors (Fig. 4J) important for differentiation which manifested in a reduction of secretory cell markers upon longer stimulation (72h) (edited statement in line 256-258).

8. Line 259, How about apoptosis gene levels here?

- Apoptosis markers BAX and BCL2 are not amongst the differentially expressed genes. (see Suppl. Table 4).

9. Line 290 : It has been shown that the response of colon explants to IFN- α was variable among donors (<https://www.sciencedirect.com/science/article/pii/S2352345X16301084#undfig1>). This study should be cited. Was the response to IFN α tested on both organoids?

- The reference was included (Line 669-671). The response to IFN α was only tested in organoid line II given the limited availability of the serum that was used for co-stimulation. Overall, however, we saw an effect in both donor lines and less response was rather connected to the serum, not the organoid donor. In the publication reported interindividual heterogeneity depends on the therewith connected release of IL-18. Future studies could

include analysis of cytokine release from the organoids after serum stimulation and a higher number of organoid lines to validate our findings.

10. Line 305-307: Is single cell sequencing from single organoid line or from combined? Do two organoid lines show different distributions?

- scRNA-seq was performed using organoid line II. Due to limited resources, we unfortunately could not include another organoid line.

Reviewer #3 (Significance (Required)):

While there is mounting evidence of an altered intestinal barrier integrity in SLE patients, there is little insights in the mechanisms. Using colon organoids is a novel approach with great potentials to investigate this issue. The strongest signature found by the authors was type IFN, which is indicates that the colon epithelial cells respond in a similar manner to other cell types. The secretory and absorption genes are of potential greater interest to unraveling mechanisms to gut alteration in SLE.

This study is of interest for audiences interested in lupus basic and clinic research, as well as investigators working of gut barrier integrity.

Thank you for your constructive feedback. We are encouraged by your acknowledgment of our novel approach using colon organoids to explore the altered intestinal barrier integrity in SLE patients. Your emphasis on the significance of the type IFN signature and the importance of secretory and absorption genes aligns with our perspective.

Incorporating the intestinal barrier functional experiment emphasizes the translational nature of our study. We agree with your view that our findings are relevant for those involved in both basic and clinical lupus research, as well as specialists in intestinal biology. Your encouraging feedback strengthens our conviction in the wider significance of our work. We are motivated to keep bridging the gap between these two essential areas of study.

References

1. Pearce, S. C. *et al.* Marked differences in tight junction composition and macromolecular permeability among different intestinal cell types. *BMC Biol.* **16**, 1–16 (2018).
2. Kishida, K., Pearce, S. C., Yu, S., Gao, N. & Ferraris, R. P. Nutrient sensing by absorptive and secretory progenies of small intestinal stem cells. *Am. J. Physiol. Liver Physiol.* **312**, G592–G605 (2017).
3. Rath, E., Moschetta, A. & Haller, D. Mitochondrial function — gatekeeper of intestinal epithelial cell homeostasis. *Nat. Rev. Gastroenterol. Hepatol.* **15**, 497–516 (2018).
4. Rodríguez-Colman, M. J. *et al.* Interplay between metabolic identities in the intestinal crypt supports stem cell function. *Nature* **543**, 424–427 (2017).
5. Ludikhuize, M. C. *et al.* Mitochondria Define Intestinal Stem Cell Differentiation Downstream of a FOXO/Notch Axis. *Cell Metab.* **32**, 889-900.e7 (2020).
6. Sünderhauf, A. *et al.* Loss of Mucosal p32/gC1qR/HABP1 Triggers Energy Deficiency and Impairs Goblet Cell Differentiation in Ulcerative Colitis. *Cmgh* **12**, 229–250 (2021).

31st Oct 2023

Dear Dr. Resnik-Docampo,

Thank you for the submission of your revised manuscript to EMBO Molecular Medicine following initial review at Review Commons. Your manuscript was sent back to the original reviewers, and we have now received their reports. As you will see below, they are supportive of publication pending minor revisions, and I am therefore pleased to inform you that we will be able to accept your manuscript once the following points will be addressed:

Referee comments:

Please address the remaining concern from referee #2 regarding barrier permeability in organoids.

Additionally, please address the following editorial issues:

- 1) Please provide a .docx formatted version of the manuscript text (including legends for main figures, EV figures and tables). Please make sure that the changes are highlighted to be clearly visible.
- 2) We can accommodate a maximum of 5 keywords, please adjust accordingly.
- 3) Please define the corresponding author(s) on the title page.
- 4) The order of the manuscript sections needs correcting to: Acknowledgments, Disclosure and competing interests statement, References, Main figure legends, EV figure legends
- 5) Please remove "Data not shown" (p.8): as per our guidelines on "Unpublished Data", the journal does not permit citation of "Data not shown". All data referred to in the paper should be displayed in the main or Expanded View figures.
- 6) Data availability section: please provide the datasets accession numbers and URL links. This section should be placed at the end of the materials and methods.
- 7) Acknowledgements: please make sure that the information provided in the submission system matches the information provided in the manuscript (currently Merck KGaA, Darmstadt, Germany is missing in the submission system).
- 8) Author contributions: CRediT has replaced the traditional author contributions section because it offers a systematic machine-readable author contributions format that allows for more effective research assessment. Please remove the Authors Contributions from the manuscript and use the free text boxes beneath each contributing author's name in our system to add specific details on the author's contribution. More information is available in our guide to authors.
- 9) Please rename the Conflict of Interest to 'Disclosure statement and competing interests': We updated our journal's competing interests policy in January 2022 and request authors to consider both actual and perceived competing interests. Please review the policy <https://www.embopress.org/competing-interests> and update your competing interests if necessary.
- 10) Please correct the reference format to alphabetical, with 10 authors listed before et al.
- 11) We can accommodate up to 5 EV figures that are collapsible/expandable online. As you currently have 7 supplementary figures, you could either merge them to have 5 figures, or compile the remaining 2 figures in an appendix, with a table of content and figure legends included in the PDF file.
- 12) The Suppl. Tables 3-5, 7-11 should be renamed Dataset EV1-8. The Suppl. Tables 1, 2 and 6 should be renamed Table EV1-3.
- 13) Please make sure that all figures are referenced in the text and in chronological order.
- 14) In the figure legends:
 - Please note that the figure legend style does not comply with the journal guidelines i.e. all the figure legends are in a run-on style.
 - Please indicate the statistical test used for data analysis in the legend of supplementary figure 2b.
 - Please note that the error bars are not defined in the legend of figures 1c, h; 2e-g; 3c-d; 4c, d, f, g, k; 5c; 7c; supplementary figures 3b, c, d; 4a, b, d; 5b; 7c.
 - Please note that information related to n is missing in the legend of figure 6c; supplementary figure 2b.
 - Please note that the box plots need to be defined in terms of minima, maxima, centre, bounds of box and whiskers, and percentile in the legend of figure 2c.
- 15) At EMBO Press we ask authors to provide source data for the main figures. Our source data coordinator will contact you to discuss which figure panels we would need source data for and will also provide you with helpful tips on how to upload and organize the files.
- 16) Please provide a complete author checklist, which you can download from our author guidelines (<https://www.embopress.org/page/journal/17574684/authorguide#submissionofrevisions>). Please insert information in the checklist that is also reflected in the manuscript. The completed author checklist will also be part of the RPF.
- 17) The paper explained: EMBO Molecular Medicine articles are accompanied by a summary of the articles to emphasize the major findings in the paper and their medical implications for the non-specialist reader. Please provide a draft summary of your article highlighting
 - the medical issue you are addressing,
 - the results obtained and
 - their clinical impact.

18) Every published paper includes a 'Synopsis' to further enhance discoverability. Synopses are displayed on the journal webpage and are freely accessible to all readers. They include a short stand first (maximum of 300 characters, including space) as well as 2-5 one-sentence bullet points that summarize the paper. Please write the bullet points to summarize the key NEW findings. They should be designed to be complementary to the abstract - i.e. not repeat the same text. We encourage inclusion of key acronyms and quantitative information (maximum of 30 words / bullet point). Please use the passive voice. Please attach these in a separate file or send them by email, we will incorporate them accordingly.

19) As part of the EMBO Publications transparent editorial process initiative (see our Editorial at <http://embomolmed.embopress.org/content/2/9/329>), EMBO Molecular Medicine will publish online a Review Process File (RPF) to accompany accepted manuscripts.

In the event of acceptance, this file will be published in conjunction with your paper and will include the anonymous referee reports, your point-by-point response and all pertinent correspondence relating to the manuscript. Let us know whether you agree with the publication of the RPF and as here, if you want to remove or not any figures from it prior to publication. Please note that the Authors checklist will be published at the end of the RPF.

I look forward to receiving your revised manuscript.

Yours sincerely,

Lise Roth

***** Reviewer's comments *****

Referee #1 (Remarks for Author):

The authors have adequately revised the manuscript and done all the experiments asked for, so in my view this manuscript should be accepted for publications

Referee #2 (Comments on Novelty/Model System for Author):

The model is adequate. Technical quality, novelty and medical impact medium. Overall a good solid paper.

Referee #2 (Remarks for Author):

I congratulate the group of Dr. Resnik-Docampo for a very elegant and convincing revision of the paper, I have only one further remark alas a very important one. While a very elegant and out of the box solution is applied for FITC-dextran permeability, I am doubtful that a monolayer provides the same quality of barrier as an organoid. Forces and mechanical stress are completely different in monolayers and organoids. I strongly suggest to show in 3d organoids not monolayers barrier permeability (<https://www.ncbi.nlm.nih.gov/pmc/articles/PMC5755602/>). This can be achieved either by microinjecting organoids or by generating inside-out organoids and exposing those to FITC-dextran. This point is crucial to me for the acceptance of the manuscript (which I evaluate positively overall) as I stated in my first revision.

Referee #3 (Comments on Novelty/Model System for Author):

This is a significant and novel study

Referee #3 (Remarks for Author):

All concerns have been addressed

Rev_Com_number: RC-2023-02065

New_manu_number: EMM-2023-18796

Corr_author: Resnik-Docampo

Title: Systemic Lupus Erythematosus Serum Stimulation of Human Intestinal Organoids Induces Barrier Leakiness and Changes in Goblet Cell Differentiation

Full Revision

Manuscript number: EMM-2023-18796 RC- 2023-02065

Corresponding author(s): Martin, Resnik Docampo

1. General Statements [optional]

This section is optional. Insert here any general statements you wish to make about the goal of the study or about the reviews.

In response to the feedback received from the editor, we have taken several steps to include all suggestions and change formatting according to the journal guidelines.

Those changes include:

- Supplementary figure legend names were changed to EV in the figure itself and in the text
- Supplementary figure 1 and 2 as well as supplementary figure 4 and 5 were merged
- Figure legends were fully revised and edited to have it more concise and to include all the information on statistical analysis, values shown and replicates
- The section of the statistical analysis was revised to include all information necessary
- The style of references was adjusted according to the style of the journal
- The section of author contributions was removed and will be uploaded in the portal
- The submission of the raw sequencing data was finalized and the reference added to the manuscript
- Title and abstract was shortened as required

For clarity:

In this document, reviewers' comments are highlighted in blue, our responses are in black. In the manuscript document the changes are marked in red except for the changes made to the reference formatting and the Supplementary figure names.

Referee #1 (Remarks for Author):

The authors have adequately revised the manuscript and done all the experiments asked for, so in my view this manuscript should be accepted for publications

We thank the reviewer again for the input and suggestions.

Referee #2 (Comments on Novelty/Model System for Author):

The model is adequate. Technical quality, novelty and medical impact medium. Overall a good solid paper.

Referee #2 (Remarks for Author):

I congratulate the group of Dr. Resnik-Docampo for a very elegant and convincing revision of the paper, I have only one further remark as a very important one. While a very elegant and out of the box solution is applied for FITC-dextran permeability, I am doubtful that a monolayer provides the same quality of barrier as an organoid. Forces and mechanical stress are completely different in monolayers and organoids. I strongly suggest to show in 3d organoids not monolayers barrier permeability (<https://www.ncbi.nlm.nih.gov/pmc/articles/PMC5755602/>). This can be achieved either by microinjecting organoids or by generating inside-out organoids and exposing those to FITC-dextran. This point is crucial to me for the acceptance of the manuscript (which I evaluate positively overall) as I stated in my first revision.

Thank you for your constructive feedback. We agree that a monolayer system differs from organoids grown in 3D. To build a relevant *in-vitro* model we use a medium that induces differentiation and together with the serum supports the presence of all major cell types. This in turn leads to a high proportion of absorptive cells. Absorptive cells express high levels of ion transporters (as also shown in our scRNA-seq data) that do not only transport ions from the apical to basal side, but also induce also the movement of water from the lumen to the basal side (Kozuka *et al*, 2017). In case of a 3D organoid where the lumen is in the center of the organoid this leads to a shrinkage of the luminal space. However, microinjection is only feasible with organoids that have high luminal pressure. Figure 1 and Figure 2 show two published protocol that show the required cystic organoid phenotype required for microinjection. In Figure 2J organoids are stated to be too differentiated to be injected. This non-cystic phenotype is comparable to our organoids stimulated with serum (Fig.3) that present with a thick epithelial layer and low internal pressure. Conclusively, we are not able to use microinjection for our serum stimulated organoids.

Fig. 1: In this protocol the cystic phenotype of the organoid is shown (Hill *et al*, 2017).

Fig. 2: In this step-by-step protocol the necessity of culturing organoids with a cystic phenotype and therewith high internal pressure is stated (Puschhof *et al*, 2021). This is achieved by using a medium that supports a proliferative phenotype with an additional stimulation of secretion (PGE2) which leads to a high internal pressure. In j organoids are stated to be too differentiated for injection. They show the lack of the cystic phenotype and are more similar to our organoids.

Fig. 3: Mophology of organoids stimulated with serum present with a phenotype that is different to the cystic organoids with a thick epithelial layer and a low internal pressure.

Organoids grown in suspension were shown to develop an inside-out phenotype which is marked by the luminal space being exposed to the medium while the basal side is facing inside (Co *et al*, 2021). This configuration would allow the addition of FITC directly to the medium, but microinjection would still be required to stimulate the organoids basally with serum. Thus, we would face a similar challenge while in addition reducing the throughput given the manual nature of the microinjection.

Full Revision

Monolayer cultures helped us to overcome these technical challenges and are additionally a well-recognized system to study barrier permeability. They are able to form a tight barrier that allows to study apical to basal translocation of molecules and even ions (van Dooremalen *et al*, 2021; Haynes *et al*, 2022, Kozuka *et al*, 2017, <https://www.stemcell.com/organoid-monolayer-barrier-permeability-assay.html>). The 2D nature of this model enabled the stimulation on the basal side with serum while having access to the luminal side for the addition of FITC. Given that the readout is not based on microscopy like in the 3D set-up where only a limited number of organoids can be analyzed, we could increase the number of technical and biological replicates using the 2D model making the read-out more robust. Until the technological burden is overcome, we are convinced that the 2D assay is the best compromise to study barrier permeability in our study where we require the access to both the luminal and basal side. Together with the transcriptional changes we can therewith conclude that SLE serum stimulation leads to alterations in the barrier that cause an increase in epithelial permeability. Future studies could explore these changes more specifically by using organ-on-a-chip approaches that support the introduction of flow or even microbes.

Referee #3 (Comments on Novelty/Model System for Author):

This is a significant and novel study

Referee #3 (Remarks for Author):

All concerns have been addressed

We thank the reviewer again for the input and suggestions.

References

- Co JY, Margalef-català M, Monack DM & Amieva MR (2021) Controlling the polarity of human gastrointestinal organoids to investigate epithelial biology and infectious diseases. *16*: 5171–5192
- van Dooremalen WTM, Derksen M, Roos JL, Higuera Barón C, Verissimo CS, Vries RGJ, Boj SF & Pourfarzad F (2021) Organoid-Derived Epithelial Monolayer: A Clinically Relevant In Vitro Model for Intestinal Barrier Function. *J Vis Exp*
- Haynes J, Palaniappan B, Tsopmegha E & Sundaram U (2022) Regulation of nutrient and electrolyte absorption in human organoid-derived intestinal epithelial cell monolayers. *Transl Res* 248: 22–35
- Hill DR, Huang S, Tsai YH, Spence JR & Young VB (2017) Real-time measurement of epithelial barrier permeability in human intestinal organoids. *J Vis Exp* 2017: 1–10
- Kozuka K, He Y, Koo-McCoy S, Kumaraswamy P, Nie B, Shaw K, Chan P, Leadbetter M, He L, Lewis JG, et al (2017) Development and Characterization of a Human and Mouse Intestinal Epithelial Cell Monolayer Platform. *Stem Cell Reports* 9: 1976–1990
- Puschhof J, Pleguezuelos-Manzano C, Martinez-Silgado A, Akkerman N, Saftien A, Boot C, de Waal A, Beumer J, Dutta D, Heo I, et al (2021) Intestinal organoid cocultures with microbes. *Nat Protoc*

4th Dec 2023

Dear Dr. Resnik-Docampo,

Thank you for submitting your revised manuscript. We have now received the report from the referee who re-reviewed your manuscript, and as you will see below, this referee is supportive of publication. I will therefore be able to accept your manuscript once the following minor editorial points will be addressed:

1/ Manuscript text:

- Please accept the previous changes in red and only keep in track changes any new modification.
- Materials and methods:
 - o Human samples: please add the complete statement that the experiments conformed to the principles set out in the WMA Declaration of Helsinki and the Department of Health and Human Services Belmont Report.
 - o Cells: please indicate whether tests for the presence of mycoplasma were performed.
 - o Statistics: please include a statement about randomization and blinding, even if no blinding was done.
 - o Please correct the checklist accordingly.
- Data Availability section: please remove "Analyzed transcriptomic data are as well provided as supplementary datasets (EV1-8). All other data supporting the findings of this study are available from the corresponding author on request."

2) At EMBO Press we ask authors to provide source data for the main figures. Please find the checklist from our source data coordinator attached and upload the requested source data accordingly.

3) Checklist:

Please make sure that the checklist reflects all information provided in the Materials and Methods; in particular, please fill in the section on Experimental study design and statistics.

4) Please note that all corresponding authors are required to supply an ORCID ID for their name upon submission of a revised manuscript. An ORCID identifier is currently missing for Inga Viktoria Hensel.

5) Thank you for providing "The paper explained". Please incorporate it in the main manuscript file.

6) Thank you for providing a synopsis text and image. I have slightly edited the text to match our style and format, please let me know if you agree with the following or amend as you see fit:

Stimulation of human intestinal organoids with serum from Systemic Lupus Erythematosus patients allowed the study of this systemic disease's impact on the epithelial barrier and gut homeostasis.

- The in vitro model contained all major intestinal epithelial cell types.
- The decrease of secretory cell population was accompanied by changes in mucus composition.
- Type I interferon signature could be inhibited by anifrolumab.
- Stimulation with serum from SLE patients led to increased barrier leakiness and altered mitochondrial function.

I look forward to reading a new revised version of your manuscript as soon as possible.

Yours sincerely,

Lise Roth

***** Reviewer's comments *****

Referee #2 (Comments on Novelty/Model System for Author):

All discussions have been made in the previous round of review and I am very satisfied with all the replies of the authors

Referee #2 (Remarks for Author):

The group of Resnik-Docampo has extensively answered all my comments. My sincerest compliments to the whole group.

Rev_Com_number: RC-2023-02065

New_manu_number: EMM-2023-18796-V2

Corr_author: Resnik-Docampo

Title: SLE Serum Induces Altered Goblet Cell Differentiation and Leakiness in Human Intestinal Organoids

All editorial and formatting issues were resolved by the authors.

5th Jan 2024

Dear Dr. Resnik-Docampo,

Thank you for submitting your revised files, and please accept my apologies for the delay in getting back to you during this busy time of the year. I am pleased to inform you that your manuscript is accepted for publication and is now being sent to our publisher to be included in the next available issue of EMBO Molecular Medicine!

If you have any questions, please do not hesitate to contact the Editorial Office.

Congratulations on your interesting work!

With kind regards,

Lise Roth

Rev_Com_number: RC-2023-02065
New_manu_number: EMM-2023-18796-V3
Corr_author: Resnik-Docampo
Title: SLE Serum Induces Altered Goblet Cell Differentiation and Leakiness in Human Intestinal Organoids